# The Future of Permanent-Magnet-Based Electric Motors: How Will Rare Earths Affect Electrification?

**DOI:** 10.3390/ma17040848

**Published:** 2024-02-09

**Authors:** Benjamin Podmiljšak, Boris Saje, Petra Jenuš, Tomaž Tomše, Spomenka Kobe, Kristina Žužek, Sašo Šturm

**Affiliations:** 1Department for Nanostructural Materials, Jožef Stefan Institute, Jamova cesta 39, 1000 Ljubljana, Slovenia; boris.saje@kolektor.com (B.S.); petra.jenus@ijs.si (P.J.); tomaz.tomse@ijs.si (T.T.); spomenka.kobe@ijs.si (S.K.); tina.zuzek@ijs.si (K.Ž.); saso.sturm@ijs.si (S.Š.); 2Kolektor d.o.o., Vojkova ulica 10, 5280 Idrija, Slovenia

**Keywords:** permanent magnets, rare-earth elements, critical raw materials, electric motor, recycling, additive manufacturing

## Abstract

In this review article, we focus on the relationship between permanent magnets and the electric motor, as this relationship has not been covered in a review paper before. With the increasing focus on battery research, other parts of the electric system have been neglected. To make electrification a smooth transition, as has been promised by governing bodies, we need to understand and improve the electric motor and its main component, the magnet. Today’s review papers cover only the engineering perspective of the electric motor or the material-science perspective of the magnetic material, but not both together, which is a crucial part of understanding the needs of electric-motor design and the possibilities that a magnet can give them. We review the road that leads to today’s state-of-the-art in electric motors and magnet design and give possible future roads to tackle the obstacles ahead and reach the goals of a fully electric transportation system. With new technologies now available, like additive manufacturing and artificial intelligence, electric motor designers have not yet exploited the possibilities the new freedom of design brings. New out-of-the-box designs will have to emerge to realize the full potential of the new technology. We also focus on the rare-earth crisis and how future price fluctuations can be avoided. Recycling plays a huge role in this, and developing a self-sustained circular economy will be critical, but the road to it is still very steep, as ongoing projects show.

## 1. Introduction

The world is shifting to combustion-free transport. New research shows that in 2021, an estimated 6.5 million electric vehicles (EVs) will be sold worldwide. Half of this number has been sold in China alone (an increase of 160% to the year 2021), which makes it the world’s largest electric vehicle (EV) market in less than a decade. Europe is heading in the same direction, selling over 2.3 million EVs in 2021, which represents 19% of total car sales in 2021 [1]. To achieve its target of net-zero greenhouse-gas emissions by 2050, set in its December 2019 “Green Deal” to transform its transport sector, a lot more EVs have to “hit the roads”. This raises the question of the raw materials needed for such an attempt [2].

Most public and scientific interest has been focused on how we will store the energy that is produced by renewable sources and how we will be able to harvest that stored energy. Batteries will probably be the main energy-storage option, although hydrogen could be a viable and possibly even better option. In either case, the efficiency of electric motors that act as converters of energy into mechanical motion will be one of the most important considerations.

All the early inventors used permanent magnets in their previously called electrical rotating machines. However, the early motors were very different from the motors of today. The first electrical motor using permanent magnets was constructed by Michael Faraday in 1821 [3]. He adopted ideas that were previously presented by Hans Christian Oersted [4] and William Wollaston [5]. Faraday’s device was very simplistic and did not look like an electric motor, but with the use of permanent magnets, a bowl of mercury, and a battery, he generated an electromagnetic field that produced mechanical motion. This triggered many new modifications to the idea, changing it to the design we know today. However, the first patent for the electric motor was not granted until 1837 to Thomas Davenport [6]. Because he used low-quality permanent magnets in his design, which produced a power output of 4.5 W, they did not sell. This made many future inventors switch to electromagnets, which were more suitable for the job at the time. Not until new types of magnetic materials, such as carbon, cobalt, and tungsten steel, were invented almost 100 years later did inventors use permanent magnets in their designs. But the real breakthrough came with the discovery of Al-Ni-Co magnets [7], where permanent magnets were able to replace electromagnets in electric motors and the development of permanent-magnet motors began.

The most efficient electric motor is a permanent-magnet synchronous motor [8]. Their efficiency makes them popular for drive motors, power steering, stop-start motors, and regenerative braking generators. These motors use permanent magnets based on rare-earth elements (REEs), in particular neodymium-iron-boron (Nd-Fe-B) and samarium-cobalt (Sm-Co), because of their high maximum energy product (*BH*)_max_ (a measure of the magnet’s performance), which is needed for the high efficiency and the high resistance to demagnetization. But there are still some challenges and gaps in their performance and application, like:Rare Earth Material Dependence:

Many high-performance permanent magnets, particularly those based on neodymium, rely on rare earth elements. The mining and processing of these materials can be environmentally damaging and subject to supply chain issues. As a result, REEs are considered by the European Commission to be the most critical raw materials in terms of their economic importance and supply risk [9]. Research is ongoing to develop alternative magnet materials that reduce dependence on rare earth elements. Another aspect is recycling, where a lot of research is conducted to improve the recyclability of permanent magnets.

Temperature Sensitivity:

Permanent magnets can lose their magnetic properties at high temperatures. This limits the operating temperature range of motors and can be a concern in applications where motors are exposed to elevated temperatures or require high-temperature resistance.

Demagnetization Risk:

Permanent magnets are susceptible to demagnetization under certain conditions, such as high temperatures or excessive magnetic fields. This can result in a loss of motor performance and reliability.

Cost of Materials:

High-performance permanent magnets can be expensive due to the cost of rare earth elements. Reducing the cost of materials while maintaining or improving performance is a key challenge in making permanent-magnet-based motors more cost-effective. For less-demanding electric motors, where size does not matter, ferrites can be used. These magnets are abundant, cheap, and have the largest share of the market.

Motor Efficiency at Partial Loads:

The efficiency of permanent-magnet motors can decrease at partial loads, which is common in many real-world applications. Improving efficiency across a wide range of operating conditions is crucial for maximizing energy savings.

Size and Weight Constraints:

In some applications, especially in industries where weight and size are critical factors (e.g., aviation and automotive), finding the right balance between power density, weight, and size remains a challenge.

Manufacturing and Integration Complexity:

Fabricating and integrating permanent magnets into motor designs can be complex. Ensuring consistent quality, especially for mass production, and addressing manufacturing challenges are areas of focus.

Durability and Long-Term Reliability:

Long-term reliability and durability are critical factors, especially in industrial and automotive applications. Researchers and engineers are continually working on improving the robustness of permanent-magnet-based motors to ensure a longer lifespan.

Dynamic Performance:

Achieving optimal dynamic performance, such as high torque density and fast response times, is an ongoing area of research to meet the demands of various applications.

Cognitive Implications:

As electric motors become more integrated into autonomous systems and artificial intelligence applications, there may be a need for motors that can adapt to changing conditions in real time. This involves developing control algorithms that optimize motor performance based on varying inputs.

In this paper, we will focus on what has been done up until now and which future technologies will help make electrification more viable, like new production technologies, recycling methods, and motor designs.

## 2. Permanent Magnets and How They Dictated History

The last 30-plus years have been dominated by Nd-Fe-B-type magnets. But for the complete picture, we must look back nearly 50 years and visit the events that led to the discovery of today’s modern metallic magnets. We will divide recent history into decades and add milestones for the triggering events (TE) that could be responsible for the subsequent discoveries (D), as shown in Figure 1.

We will look at the following:TE: “Pile of Samarium in the backyard“ at the beginning of the 1960s;D: Sm-Co (SmCo_5_, Sm_2_Co_17_) in the mid-1960s and early 1970s;TE: Cobalt crisis, which occurred at the end of the 1970s;D: Nd-Fe-B development activities at the beginning of the 1980s;TE: EC Research Crisis: We have to do something!, which happened in 1985 and resulted in CEAM;D: Sm-Fe-N, which appeared at the very end of the 1980s and 1990s;TE: Rare Earth crisis, which we all still remember from 2011;D: Today’s new materials and technologies with their perspectives.

To be fair to non-metallic, but still hard, magnetic materials, around 80% by weight of today’s permanent magnets are still sintered hard ferrites, as shown in Figure 2 left, and 70% of this ends up in various motors. On the other hand, around 65% of the market value is Nd-Fe-B (Figure 2 right), and again, about 70% of this ends up in motors of all kinds.

We will concentrate more on RE-TM magnets, where RE stands for Nd and Sm and TM for Fe and Co, and not so much on ferrites. Before the discovery of Nd-Fe-B magnets in 1984 [11,12], the major players were ferrites, Sm-Co (50 wt.% Co), and Alnico (15 wt.% Ni, 30–35 wt.% Co) magnets. For demanding, high-temperature applications, Sm-Co magnets were dominant (high-performance, high-temperature, defense-related uses). They were developed in Dayton, USA, by the group of K. Strnat [13], who was originally working on the Y-Co system. As recalled by Alden Ray [14], the triggering event was “There was quite some unused Sm pilling up in a backyard, so one day we thought, let us mix it with Co and see what happens”. Eventually, a new compound was discovered—firstly SmCo_5_; followed by Sm_2_Co_17_; both of which proved to be successful [15]. Raw-material prices and availability were of no particular concern then, but at the end of the 1970s, the so-called cobalt crisis occurred and then reappeared in 2008 and 2018 (as shown in Figure 3—collected from references [16,17,18]).

Supplies of Sm were of no concern, but magnet producers were closely monitoring the Co and Ni markets. Some claim this was the trigger for the development activities on non-cobalt magnets, which resulted in the pioneering work on Nd-Fe-B at the US Naval Labs. At the beginning of the 1980s, N. Koon observed anisotropy in the Tb-La-Fe system that was not as expected [19]. Shortly after, M. Sagawa discovered sintered, anisotropic Nd-Fe-B while working at the Sumitomo Special Metals Company [11], and J.J. Croat and his group [20] discovered isotropic melt-spun ribbons and polymer-bonded magnets at General Motors. These latter materials and technologies still dominate REE-based bonded magnets [12]. Nd-Fe-B magnets have the highest maximum energy product (*BH*)_max_ at room temperature of over 400 kJ/m^3^ (Figure 4) [11].

The material itself is possible to describe as follows:Nd-Fe-B = Nd_2_Fe_14_B + Nd-rich + NdFe_4_B_4_
where the Nd_2_Fe_14_B phase is the carrier of the magnetic properties, the Nd-rich compound enables sintering, and the NdFe_4_B_4_ is difficult to get rid of for thermodynamic reasons. The RE-Fe-B system has a useful property that leads to RE interchangeability and a whole range of magnetic properties. Figure 5a shows the dependence of saturation magnetization (the potential for high remanence in a magnet) for different REs used in RE_2_Fe_14_B [21]. We can observe something similar for the magnetocrystalline anisotropy (the basis for high coercivity in the magnet), as shown in Figure 5b. With this interchangeability, we can modify the magnetic properties to fulfill the needs of the application.

The discovery of Nd-Fe-B in 1984 led to »shock and awe« when news of this new family of magnetic materials was brought to the European magnetics research community, signaling that researchers in the USA and Japan had effectively pulled ahead [22]. European researchers reacted with a program called the Concerted European Action on Magnets (CEAM), which some claim was the triggering event that led to the discovery of the Sm-Fe-N magnetic phase. In the autumn of 1989, Iriyama and Kobayashi of Asahi Kasei patented a new magnetic compound—Sm-Fe-N [23]; which forms when the Sm_2_Fe_17_ binary alloy turns its in-plane anisotropy to uniaxial as nitrogen begins to occupy the interstitial sites in the crystal lattice. Shortly after, H. Sun and M. Coey published a similar result, and this happened within the CEAM program [24]. Sm-Fe-N has magnetic properties very similar to those of Nd-Fe-B at room temperature, with a 150 °C higher Curie temperature. Its major drawback is that it decomposes at around 550 °C, making it suitable mostly for polymer-bonded magnets.

What we today call the rare-earths crisis began at the end of 2010 and reached a peak in March 2011, lasting until August 2011 [25]. It resulted in sharp price rises, especially for the heavy rare earths (HREs) (as shown in Figure 6 for selected REs), and raised many questions about affordability, availability, and the potential monopolization of resources by China.

Today we are at pre-crisis price levels, and post-festum, we can have a closer look at those fears from 5 years ago:There will be a lack of REs: So far, it has not happened because demand has not followed the predictions. The growth that was foreseen in 2009 has not materialized yet;Chinese mining, production, and export quotas: actually never influenced the shortage of material;Shipments failed: the shipment period was longer (particularly between China and Japan), but they never actually failed;China as the only source: If in 2011, 95–97% of Nd for magnets were coming out of China, today (2024), the figure is only 80;Predictions in other fields (phosphor lighting tubes and LEDs—Rare Earth Conference; Shenzhen; 2013)—due to the transition from phosphors to LED; phosphor applications have seen a drop in the consumption of Res; which are then used elsewhere.

A positive outcome of the RE crisis is that it triggered the opening of new and abandoned mines to reduce the dependence on China. Even though the production of rare earth metals will rise to 300.000 metric tons worldwide in 2022, that is up significantly from 190,000 MT in 2018, just four years prior, with 70% still coming from China, but other countries are ramping up their production (Figure 7).

It also opened up new research fields as to how to reduce dependence on rare earths:Less critical REs and heavy REs, such as Dy and Tb;Recycle REs from end-of-life goods and magnets;New magnetic materials (possibly with no REs);RE-free magnets.
which we will look at more closely in the following section.

### 2.1. Resource-Efficient Nd-Fe-B PMs with Fewer REEs and/or Fewer HREEs

The high operating temperatures of the traction motors for electric vehicles, i.e., approximately 200 °C, mean that Nd–Fe–B ternary magnets cannot be used due to their relatively low Curie temperature (310 °C) and the negative temperature dependence of coercivity. It was calculated that ~2400 kA/m of coercivity at room temperature would be enough to have an appropriate coercivity left that could be used for motor applications [27]. Here heavy rare-earth elements come into play, as in theory, the (Dy/Tb,Nd)_2_Fe_14_B phase could reach this value, where we substitute approximately one-third of the Nd atoms with Dy or Tb. They help to reach a higher magneto-crystalline anisotropy in this phase [28]. Applications that contain Dy via traditional alloying are high-temperature motors and generators, hybrid and electric traction drives (up to 11 wt%), commercial and industrial generators up to 6.5 wt%, e-bikes, energy storage systems, magnetically driven transportation, motors, wind-power generators up to 4.2 wt% hard disk drives, and MRI devices, which contain up to 1.4 wt% of Dy [29]. However, if we use conventional powder metallurgy, Dy and Tb antiferromagnetic cations couple with Fe. This reduces the magnetization and thus the energy product because it has the most dominant effect on it. This can be overcome with the grain-boundary diffusion process (GBDP), where Dy and Tb are used only locally [30,31].

The technology of grain surface HRE deployment in magnet microstructure has a different name with different producers, for example:Binary Alloys Method (Grain Boundary Diffusion Process-GBDP-Shin Etsu);High Anisotropy Field Layer (HAL-TDK);DD Magic™ (Deposition and Diffusion-Hitachi).

To achieve those local dispersions of Dy and Tb, their ceramic precursors are used on top of already sintered magnets. These small additions are very much in line with the scarcity of HREEs like Dy and Tb. What is happening is that the heavy rare earths are then diffused into the magnet and substitute Nd in the matrix phase, producing a so-called core-shell structure, which prevents easy demagnetization when exposed to an external reverse magnetic field. By adding less than 1 wt.% Dy, the coercivity can be increased by more than 25%, which is shown in Figure 8 [32]. This was an outcome of the FP7 European project ROMEO (Replacement and Original Magnet Engineering Options), where an innovative approach to minimizing the amount of HRE needed was developed.

Wang et al. report the highest coercivity by separately diffusing Dy_70_Cu_30_ and then Pr_68_Cu_32_ eutectic alloys through grain boundaries in fine-grained Dy-free sintered magnets. They used a two-step GBD process, which exhibited a coercivity of 2230 kA/m and an excellent temperature coefficient of coercivity of β = −0.447%/°C. During the second Grain Boundary Diffusion (GBD) process, the infiltration of Pr–Cu eutectic markedly enhanced the distribution, thickness, and chemical composition of the essential Dy-rich shell [33].

Because Praseodymium is chemically very similar to Nd, it is used to substitute it in Nd-Fe-B alloys to reduce costs. As it also does not lead to the formation of additional phases, in contrast to other light RE elements such as La [34,35,36], Ce [37,38,39], and Y, it can fully substitute Nd. The problem with a lower Curie temperature is counterbalanced with the addition of Co. A systematic investigation of the Pr-rich Pr-(Fe,Co)-B material system was carried out by Wu et al. [40]. Pr doping is known to increase the coercivity of Nd-Fe-B permanent magnets, but no in-depth study has been conducted on the impact of Pr on phase formation and microstructure. They investigate the phase formation, microstructure, magnetic properties, and coercivity mechanisms in Pr-rich Pr_15_(Fe_1−x_Co_x_)_78_B_7_ (x = 0 − 1) alloys, where x = 0 and 0.2 produce a Pr-rich phase that hinders reversal magnetization, resulting in a high coercivity of 1400~1600 kA/m with good thermal stability. Micromagnetic modeling reveals that the change in the direction of magnetization at the interfaces of exchange-coupled grains often has a very complex topology, and the transition regions between two interaction domains are “pinned” at the structural inhomogeneities.

Ce and La are also interesting in substituting Nd in the main alloy, as they represent about 70% of the total amount of rare earths in minerals that are excavated and are largely unused and stored. Substitution of Nd by them reduces the performance of the Nd-Fe-B hard magnetic compound, mainly due to the lower intrinsic properties. Delette presented a good overview of substituting with non-critical light rare earth elements [41]. Regarding the Ce and La, it has been demonstrated that an excess in rare earth content and the Co addition limit the formation of some detrimental secondary phases. Coercivity values higher than 1430 kA/m have been achieved for a cerium substitution rate of 20%. These performances correspond to RE contents close to 31% wt. [42]. Co improves the Curie temperature of Ce-substituted magnets and the energy product of melt-spun ribbons from 85 kJ/m^3^ to 128 kJ/m^3^ in (Nd_1−x_ Ce_x_)_2+y_Fe_14-z_Co_z_B with z = 2 [43]. The effectiveness of coercivity in magnets substituted with Ce and La has seen significant improvements through grain boundary engineering techniques. These methods primarily involve creating intergranular secondary phases that are non-ferromagnetic and dispersed around the hard phase. This strategy aids in facilitating magnetic exchange and decoupling among neighboring grains. Techniques such as Infiltration, Grain Boundary Restructuring (GBR), and Grain Boundary Diffusion Process are key examples of this approach. With GBR, an additional powder with a high RE content and/or a low melting temperature is mixed with the RE-Fe-B powder before sintering. The gain in performance with the GBR method is generally lower than the one provided by the Grain Boundary Diffusion Process [44]. These types of magnets are already on the market, as the Inspires project [45] has shown when examining EOL magnets. They are used as lower-quality magnets containing up to 8% cerium in electric scooters. This raises the question of whether this can be as easily recycled as “pure” Nd-Fe-B magnets.

### 2.2. Recourse-Efficient Nd-Fe-B PMs via Recycling

The recycling of Nd–Fe–B permanent magnets (mainly sintered) from EOL products is nowadays categorized as a “key enabling technology” for positioning REs within the circular economy. Sintered Nd–Fe–B PMs are really multi-phase materials. The hard magnetic phase represents only 85–87% of the whole magnet; the rest are Nd-rich grain-boundary phases (GBPs) that also contain Nd-oxide phases [46]. Today, there are different techniques available to recycle Nd-Fe-B magnets. They can be divided into reusing, reprocessing, and remelting. The most economically viable route is the reuse of EOL magnets without changing the chemical structure of the magnet. This is only used for generators and motors in wind turbines and EV/HEVs. A mechanical dismantling and separation technique was developed by the Hitachi Group (Japan). They used a rotational drum to separate Nd-Fe-B magnets from hard disk drives (HDDs). Another technique they developed is recycling magnets from air conditioner compressors, which are cut off directly and further demagnetized thermally at 400–500 °C or by resonance damping demagnetization (Figure 9) [47,48].

However, this process is very limited as the magnets, after salvaging them, already have a defined shape and properties that are useful only in the same applications. The more common route today is direct re-use via remelting the magnets without generating any waste. This is mostly used by magnet producers to put internal scrap back into the production line. However, this becomes more complicated for EOL magnets in waste electronic equipment (WEE), for example. The recycling rate of permanent magnets is very low because they are too small to be removed from WEE and are just remelted with ferrous or nonferrous [49]. The issue is also that Nd–Fe–B magnet scrap has a higher oxygen content (typically 2000–5000 ppm) compared to virgin magnets (typically 300–400 ppm) [50]. This additional oxygen is mainly trapped in the REE-rich materials; e.g., this is a huge problem as they do not sinter at the low temperatures of regular magnet sintering. This produces low-density parts with low hard magnetic values [51,52]. Additions of either several wt.% NdH_2_ [53] to the recycled powder or extra Dy [54] are thus needed to induce an increase in the coercivity to the original values. However, if the oxide phase could somehow be removed prior to or during recycling, the addition of extra REEs could be avoided. One way of tackling this problem was suggested by Xu et al. [55]. Using a novel selective electrochemical recycling concept, they could recover pure matrix Nd_2_Fe_14_B grains. This presented a very environment-friendly way to remove the unwanted oxide phases directly with a low-energy input. There are other methods of REE recycling. To obtain master alloys, one would use the high-energy-demanding pyrometallurgical route, which is useful only for highly concentrated magnet scrap [56,57,58]. Leaching is also a successful way to recycle these materials using H_2_SO_4_ [59] or HCl [60]. Solvent extraction [61,62,63], ion exchange, or ionic-liquid techniques [64,65,66,67] are used to get REE species from it. The final product is achieved by selective precipitation and conversion to REE fluorides or oxides. But this technique is very environmentally problematic because of the large amounts of chemicals used, not to mention the large amount of wastewater that is generated.

Another technique to separate the magnet from its environment was developed by the University of Birmingham. The HPMS (Hydrogen Processing of Magnet Scrap) was developed to retrieve environmentally friendly Nd-Fe-B powder environmentally friendly from end-of-life magnets [68,69]. As Nd-Fe-B magnets break down into a friable, demagnetized, hydrogenated powder that can be separated mechanically from the remaining impurities, like the nickel coating, by, e.g., tumbling electronics in a rotating hydrogen reactor and collecting the powder, the magnet-free component can now be recycled much more easily. This procedure is used by the company Hypromag Ltd., Birmingham, UK to develop a full recycling supply chain for rare-earth magnets. Figure 10 illustrates how the HPMS process pulverizes the Nd-Fe-B magnet of a hard disk drive.

Because of the difference in composition [71], it is not so easy to use the recycled powders directly in the normal production line by just mixing them with a fresh powder. Sometimes additional REEs have to be added to compensate for the loss during their lifetime. This can also alter the end properties of the magnet. To avoid these problems, a more standardized production method should be implemented, as suggested by Ueberschaar et al. [72]. These problems are also tackled in the ongoing ERA-Min 2 project MaXycle, which will classify EOL magnets for recyclability in a standardized grading system and develop a labeling system for newly produced magnets for easy recyclability at their end of life [73]. By investigating the EOL magnets from different fields, it will also give recommendations for a more recycle-friendly design for dismantling and suggest new coatings for easy removal. To make these recycling routes viable, large-scale production has to lower the prices for these kinds of recycled magnets. Currently, four pilot plants are being constructed in Europe in the scope of the Susmagpro project to increase the output of recycled magnets via the HPMS route [74]. They are developing automated sensing and robotic sorting lines to identify, sort, and extract Nd-Fe-B magnets more efficiently. They will use the HPMS method to harvest the magnetic powder, which will be used in four different reprocessing routes and will be installed by partners in electric motors, water pumps, loudspeakers, and headphones. Although the core technology for creating powder from recycled magnets is mature and ready for deployment, certain segments of the value chain still require refinement to fully realize a genuinely cost-effective circular economy approach [75].

### 2.3. Non-REE-Based PMs

High-energy magnets without REEs have not been produced. To use PMs without REEs, we can choose among ferrites, alnico, and Fe-Co-Cr, but they do not match the properties that REE-based magnets can achieve. Because of the REE crisis, research in the EU was focused on finding alternatives to low-performance REE PMs in the intermediate energy-product range between 50 and 200 kJ/m^3^ [76,77,78,79]. Novel, low-cost hybrid magnets based on ferrites/alnicos, or any of their combinations represent an interesting and viable solution to close the above-mentioned gap in magnetic performance between ferrites and RE magnets. A promising development for low-temperature applications was doping ferrites with B_2_O_3_, affecting the temperature coefficient of coercivity, which, unexpectedly, switched sign, enhancing the effect of the temperature on the magnetic properties (Figure 11) [80]. Guzmán-Mínguez et al. improved the magnetic performance of strontium ferrite sintered magnets using silica by tuning sintering parameters [81].

Exchange coupling between the hard and soft magnetic phases is one of the magnetic interactions that could lead to an increase in (*BH*)_max_ [82,83]. The latter tends to increase with an increase in the coercivity, *H*_cJ_ (the contribution of the hard phase), and/or the saturation magnetization, *M*_S_ (the contribution of the soft phase) [84,85]. Therefore, by choosing wisely the constituent phases of the hybrid magnet, the energy product can be increased beyond the energy product of the hard phase alone [83]. Although this seems like an elegant solution, several requirements and restrictions have to be considered for an effective exchange. First, the constituent phases have to be in close contact, e.g., in the form of a core-shell structure [86,87,88], a layered structure [89,90,91], or as particulate composites [92,93,94,95]. Second, the particle size of the soft phase is limited by the selected hard phase. So, to effectively exchange the hard and soft magnetic phases, the particle size of the soft phase should not exceed twice that of the domain-wall width of the hard-phase material [83,88]. Some recent studies even indicated that for an efficient exchange coupling, the phases also need to have some degree of structural matching [96,97]. Furthermore, to increase the (*BH*)_max_ of the composite as much as possible, the particle size of the hard phase should be at the limit of the single-domain particle size (to obtain the maximum coercivity). This means that for Sr-ferrite (or Ba-ferrite), if used in hard-soft magnetic composites, the preferred particle size would be around 1 µm, and the particle size of the soft-phase material should not exceed 28 nm [98]. The majority of the published research in the field of exchange-coupled composites is on powders only [95,99,100,101], and papers reporting increased (*BH*)_max_ values based on exchange-coupled hard-soft bulk magnetic materials are scarce. Nevertheless, a (*BH*)_max_ of 14.3 kJ/m^3^ for a hard-soft magnetic ceramic composite was reported by Debangsu Roy and P. S. Anil Kumar [102], in which they presented a hard-soft magnetic composite consisting of BaCa_2_Fe_16_O_27_ as the hard phase and Fe_3_O_4_ as the soft phase. Torkian and Ghasemi [103] presented SrFe_10_Al_2_O_19_/(x)Co_0.8_Ni_0.2_Fe_2_O_4_ composites with a (*BH*)_max_ of 29.5 kJ/m^3^. During two European projects (the FP7 project Nanopyme [104] (GA 310516) and the H2020 project AMPHIBIAN [105] (GA 720853)), some encouraging results on sintered hybrid composites were obtained. Sr-ferrite/Co-ferrite bulk composites with a (*BH*)_max_ of 26.1 kJ/m^3^ (a schematic and the actual prepared sample are presented in Figure 12) and hard-soft Sr-ferrite/(Mn, Zn)-ferrite hybrid magnets with a (*BH*)_max_ of 23.7 kJ/m^3^ were prepared. A lot of efforts have been devoted to the research of modified or new magnetic materials that would fill the gap between REEs-based PMs and ferrites, but the vast combination of possible candidate materials, the preparation and consolidation conditions, and the fact that some options are viable only in theory make this task a highly demanding one, and so the quest to find a new magnet with improved properties continues.

### 2.4. New Permanent Magnetic Materials

In the past 10 years, a lot of scientific effort has been applied to develop a new magnetic material that would overcome the problems associated with REEs and have sufficient magnetic properties. Developing a new material to challenge anisotropic Nd–Fe–B magnets is very demanding. To replace Nd_2_Fe_14_B with a new hard magnetic compound, we need a material with a higher *M*_s_ (*μ*_0_*M*_s_ > 1.6 T) and a sufficient magnetic hardness of *H*_a_ > 1.35 Ms. Some of the promising results are summarized in Figure 13, where alternatives to REE-containing compounds are presented. To have superior magnetic properties, we want to be in the upper right-hand corner of the graph. If we take Nd_2_Fe_14_B as a reference point, we can see that REEFe_12_ compounds with the ThMn_12_ structure are very interesting and have been extensively studied as a potential high-performance permanent-magnet material. Since the molar fraction of Fe in the REE-Fe_12_ compounds is the highest among various REE_(m−n)_Fe_(5m+2n)_ compounds, a high spontaneous magnetization *μ*_0_*M*_S_ is expected for REE-Fe_12_. However, most of the REE-Fe12 binary compounds are not thermodynamically stable. To obtain the REE-Fe_12_ phase in bulk form, the Fe must be substituted with a stabilizing element M, such as Al, Cr, V, Ti, Mo, W, Si, and Nb, which reduces the *μ*_0_*M*_S_ [107]. If the REEFe_12_(N) phase could be synthesized without M, it is clear that NdFe_12_N is a very promising magnetic material—it has a higher anisotropy field and a higher magnetization (see Figure 14a) than Nd_2_Fe_14_B. Unfortunately, it decomposes at elevated temperatures (see Figure 14b, just as Sm-Fe-N does), which makes it suitable only for bonded applications [108]. Unlike NdFe_12_, SmFe_12_ shows a large uniaxial anisotropy without nitrogenation. Hirayama et al. [108] found that Sm(Fe_0.8_Co_0.2_)_12_ has excellent intrinsic hard-magnetic properties as a thin film by doping it with 3.7 at.% of B, which achieved a spontaneous magnetization of 1.78 T and a Curie temperature of 859 K. H. Sepehri-Amin et al. [109] achieved a high remanent magnetization of 1.50 T simultaneously with a high coercivity of 950 kA/m. Mn-based compounds have also been intensively studied, but compounds such as MnBi [110] and MnAl(C) [111] have only a small *μ*_0_*M*_S_ of ∼0.7 T. Recent studies by Jia et al. [112] showed that a twin-free microstructure that negatively affects the magnetic properties of conventionally fabricated L1_0_-type and RETM_12_ can be suppressed by particle sizes below a critical size of *D*_t_~300 nm. The problem with all these mentioned materials—Fe_16_N_2_ particles [113]; tetragonal-FeCo thin films epitaxially grown on substrates [114,115]; and L1_0_-FeNi thin films [116]—is the production in bulk form. Production methods using these types of nano-sized materials, either by bonding or sintering techniques, have to be developed [112].

The current status of developments can be summarized as follows:With new materials, we are achieving a (*BH*)_max_ of around 100 kJ/m^3^);This is better than ferrite (40 kJ/m^3^) and isotropically bonded Nd-Fe-B (40–80 kJ/m^3^);It is on par with anisotropically bonded Nd-Fe-B and Sm-Fe-N.

Unfortunately, it is still far from sintered Nd-Fe-B or Sm-Co.

## 3. Permanent-Magnet-Based Electric Motors

The electric motor has evolved over the years to become irreplaceable in many areas of industry. Electromagnetic induction is the basic principle on which an electric motor operates. Magnetic and electrical energies create an electromotive force in a closed circuit that conducts a current. Electric motors are classified into three major categories: direct current (DC) motors, alternating current (AC) motors, and special purpose motors [117]. While both AC and DC motors serve the same function of converting electrical energy into mechanical energy, they are powered, constructed, and controlled differently [118]. The fundamental difference is the power source. AC motors are powered by alternating current (AC), while DC motors are powered by direct current (DC), such as batteries, DC power supplies, or an AC-to-DC power converter. DC wound field motors are constructed with brushes and a commutator, which add to the maintenance, limit the speed, and usually reduce the life expectancy of the motors. AC induction motors do not use brushes; they are very rugged and have long life expectancies. The final basic difference is speed control. The speed of a DC motor is controlled by varying the armature winding’s current, while the speed of an AC motor is controlled by varying the frequency, which is commonly conducted with an adjustable-frequency drive control. The decision about an AC or DC motor system is dependent on the application and the costs. Even though AC motors have low maintenance costs and a low power demand on start-up, they do have quite some drawbacks compared to DC motors. They are harder to control than DC motors, have lower starting torque, slow starting and stopping, and no reversing, and DC motors can vary speeds by changing the voltage. While the market for AC motors is larger than that for DC, DC technology is more cost-effective than AC for lower-horsepower applications. Both have numerous different design options. They all work on roughly the same principle. The rotor and stator are located inside the cylindrical groove. The rotation of the rotor is excited by a magnetic field that repels its poles from the stator (fixed winding). This magnetic field can be maintained by reconnecting the rotor windings or by forming a rotating magnetic field directly in the stator. The first method is inherent in collector electric motors, and the second is asynchronous three-phase [119]. An overview of AC and DC motors can be seen in Figure 15. It should provide a concise and precise description of the experimental results, their interpretation, and the experimental conclusions that can be drawn.

The two most common motors that use PMs are the BLDC motor, also called the brushless motor, and the permanent-magnet synchronous motor (PMSM) (Figure 16). The brushless DC and PMSMs consist of a permanent magnet, which rotates (the rotor), surrounded by three equally spaced windings, which are fixed (the stator). The current flow in each winding produces a magnetic field vector, which sums with the fields from the other windings. By controlling the currents in the three windings, a magnetic field of arbitrary direction and magnitude can be produced by the stator. Torque is then produced by the attraction or repulsion between this net stator field and the magnetic field of the rotor [120]. The difference between them is the winding of the coils, which gives different drive signals. A PMSM is driven sinusoidally, while a BLDC is driven trapezoidally, making the PMSM much quieter, both electrically and mechanically. Plus, it has virtually no torque ripple. A brushless DC motor is an upgraded version of the brushed DC motor. The absence of brushes gives BLDC motors the ability to rotate at high speed and with increased efficiency compared to brushed DC motors, as they are easier to maintain. By varying the current flowing through the stator, the speed of the motor can be varied.

Advantages of BLDC Motors [121]:Durability;Efficiency of almost 85–90%;Ability to respond to the control mechanisms at high speeds;No sparks and less noise, as the brushes are absent;Ease of motor control (using BLDC motor-controller solutions);Ability to self-start;Cooled by conduction and requires no additional cooling mechanism.

Advantages of PMSMs:Higher efficiency than brushless DC motors;No torque ripple when the motor is commutated;Higher torque and better performance;More reliable and less noisy than other asynchronous motors;High performance at both high and low speeds of operation;Low rotor inertia makes it easy to control;Efficient dissipation of heat;Reduced size of the motor.

Both have the same disadvantage, which is the scarcity and the relatively high prices of the REEs used in them. Because of that, they tend to be more expensive than other solutions, their operating temperature is limited, and the demagnetization possibility limits the input current.

Another aspect is the positioning of the magnets. The magnets can be arranged either on the surface of the rotor (surface permanent-magnet (SPM) motors) or embedded in the rotor (interior permanent-magnet (IPM) motors) (Figure 17). SPM motors are relatively simple to understand; however, IPMs can be a little more complex. The rotor will be made of a ferrous material and used to concentrate the magnetic flux by cutting slots in it to create a flux path. The magnets are typically arranged in a V configuration, which allows the field to be concentrated, making for a stronger, more concentrated magnetic field than would be possible with a surface magnet machine [122]. Because of this, it consumes up to 30% less power, can respond to high-speed motor rotation by controlling the two types of torque using vector control, and since the permanent magnet is embedded, mechanical safety is improved as the magnet will not detach due to centrifugal force [123]. But because of their complex design, they are also more expensive.

The magnets can also be attached to the rotor, as shown in Figure 18. These motors have an external rotor radial-flux construction, compared to the conventional internal rotor radial-flux construction, and the magnets are mounted on the interior surface of the motor, typically bonded in place. The rotor spins outside the stator, which is fixed inside the rotor. They provide more torque at lower RPM due to the improved mechanical advantage offered by the rotor/stator configuration of the active magnetic configuration when compared to the internal rotor design. The weakness of this configuration is the difficulty of cooling the stator and rotor because the stator is inside the machine with no thermal pathway. They are also more difficult to seal from external conditions because of the rotor configuration, and they have relatively high inertia [124].

As an alternative to conventional radial flux motors (RFMs), axial flux motors (AFMs) have been the subject of numerous research studies. The RF structure is the common one, with an external cylindrical stator and an internal cylindrical rotor. This RFM is widely used in industrial applications; thus, it is considered the reference solution. Images of the radial flux permanent magnet synchronous machine (RF-PMSM) and the axial flux permanent magnet synchronous machine (AF-PMSM) are presented in Figure 19. Because of the discovery of new materials, improvements in manufacturing technology, and innovation, AF-PMSMs are increasingly recognized as having better power density than RF-PMSMs and being more compact [126,127]. In addition, they have better ventilation and cooling arrangements. Moreover, AF-PMSMs offer a higher torque-to-weight ratio due to the application of less core material, a smaller size, a planar and easily adjustable air gap, lower noise, and lower vibration, which make them superior to radial flux machines [128,129,130]. The problem with the AFM is the strong axial magnetic attraction between the stator and the rotor, which causes the deflection of rotor discs and fabrication difficulties such as laminations in the slotted stator, high cost, and assembly [131].

An important part of the PM motors are the soft magnetic materials that work as the motor iron cores. Soft iron cores play a crucial role in electric motors due to their magnetic properties. They are used in the windings of electric motors to amplify the magnetic field produced by the current. Soft magnets have high magnetic permeability, meaning they easily become magnetized and enhance the strength of the magnetic field. This is crucial for the motor’s efficiency and performance. They also help in reducing energy losses due to their low coercivity. Since soft magnets have low coercivity, they can quickly gain and lose magnetization with minimal hysteresis loss (energy loss due to the lag between magnetization and the magnetic field). This property makes it ideal for the alternating magnetic fields in motor applications. By increasing the magnetic field strength, soft magnet cores contribute to a more powerful interaction between the stator (the stationary part of the motor) and the rotor (the rotating part of the motor). This interaction results in higher torque and improves the efficiency of the motor. In electric motors, eddy currents (circular electric currents induced within conductors by a changing magnetic field) can cause energy loss and heating. Soft magnet cores are often laminated (composed of thin layers insulated from each other) to minimize eddy currents. The lamination breaks up the path of these currents, reducing their intensity and the associated energy loss. Most of all, they provide a path with low reluctance (opposition to magnetic field flow) for the magnetic flux. This helps in concentrating the magnetic flux in the desired area of the motor, which is essential for effective motor operation (Figure 20).

The most commonly used materials for soft iron cores in permanent magnet electric motors include:Electrical Steel (Silicon Steel): Preferred for its high magnetic permeability and low core loss, ideal for reducing eddy current losses. Commonly used in stator and rotor cores [133,134].Soft Ferrites: Used in smaller, high-frequency motors like brushless DC and stepper motors, offering high magnetic permeability and minimal eddy current losses [135].Iron-Cobalt Alloys (e.g., Hiperco): Suitable for high-performance motors requiring high magnetic saturation and flux density [136].Amorphous and Nanocrystalline Metals: Chosen for high-efficiency motors due to their extremely low hysteresis and eddy current losses and high magnetic permeability [137,138].Soft magnetic composite (SMC): SMC is a type of material that is made by mixing fine particles of iron with a polymer binder. SMC is often used in motors that require a lightweight and compact core [139].

The choice depends on the motor’s size, frequency, efficiency, and cost considerations, with electrical steel being the most common for its balance of properties and cost-effectiveness. In today’s motors, Silicon steel is the most commonly used material in laminated form. Lab studies show that adding up to 6.7% silicon to iron-silicon (Fe-Si) alloys enhances their soft magnetic properties and reduces power losses during magnetization [134]. However, silicon content above 3.5% reduces the ductility of these alloys, and above 5.0%, it causes embrittlement, hindering hot and cold workability. As a result, most industrial-scale Fe-Si alloys contain less than 3.5% silicon, as higher silicon levels complicate the conventional production of electrical steel sheets [140]. A lot of research has been going on to solve this problem with additive manufacturing techniques like laser-based AM techniques, including selective laser melting, powder bed fusion of metal with a laser beam, binder jet printing, and direct energy deposition [141,142,143,144,145,146,147]. Research is also ongoing in multi-material AM. Different multi-material powder bed fusion techniques are being developed for different applications [148]. The idea is the development of a delivery system where accurate depositions of small amounts of powders achieve the formation of multi-material intra-layers, making it possible to print soft magnetic housings and rotors of electric motors and introduce insulation materials inside the Fe-Si alloys. It can potentially improve the power-to-weight ratio of electric motors by using different materials and simplifying production without using embedding devices [149].

### 3.1. Alternatives

#### 3.1.1. Three-Phase AC Induction Motors

In the majority of cases, an induction motor is the most modest electrical machine from the construction point of view. The motor works on the principle of induction, where an electromagnetic field is induced in the rotor when the rotating magnetic field of the stator cuts the stationary rotor. Induction machines are by far the most common type of motor used in industrial, commercial, or residential settings. Its characteristic features are [150,151]:Simple and rugged constructionLow cost and minimum maintenanceHigh dependability and sufficiently high proficiencyNeeds no additional starter motorThey are naturally de-excited in the case of an inverter fault, and this is very welcome among car manufacturers.

The induction machines, however, are less efficient, larger, and heavier than BLDC motors, produce a lot of heat, require a complex inverter circuit, and are difficult to control [152,153].

#### 3.1.2. Switched Reluctance Motor (SRM)

More concern was recently directed to the SRM, due to the unequivocal points of interest of its straightforward and tough configuration, fault-tolerant operation, straightforward control, and exceptional torque-speed characteristics. An SRM can run with a consistent power range [154,155]. An SRM has several inconveniences, such as acoustic commotion descent, torque ripple, unique converter topology, above-the-top transport current ripple, and electromagnetic interference (EMI) [156,157].

#### 3.1.3. Synchronous Reluctance Motor (SynRM)

Synchronous reluctance motors emerge as one of the most promising solutions capable of meeting various requirements while ensuring high efficiency and cost-effectiveness. The key to their advantages and limitations lies in their rotor structure, which consists of a meticulously cut stack of laminations—distinct from wound rotor machines with excitation coils, squirrel cage induction machines (SCIMs) with short-circuited conductors, or permanent magnet machines with magnets. This results in a cost-effective design that leverages the reluctance principle to generate torque.

Compared to switched reluctance motors, synchronous reluctance motors exhibit significantly smaller torque ripples, and their efficiency surpasses that of SRMs. Additionally, owing to the lower phase current in synchronous reluctance motors, their inverters or power electronics come at a reduced cost. However, a notable drawback is the inability to control speed due to its constant speed application. Efforts are underway to devise new concepts addressing this limitation. A comparison is presented in Figure 21 [158].

### 3.2. New Concepts

Nowadays, electric vehicles use single-speed reduction boxes designed to let the electric motors rotate at a high, efficient RPM while the wheels spin more slowly. Such a system is heavy, complex, and expensive. Linear Labs is developing the Hunstable Electric Turbine (HET) [160]. The HET is a three-dimensional, circumferential flux, exterior permanent-magnet electric motor that runs four rotors, whereas other motors typically run one or two (Figure 22). The stator is fully encapsulated in a four-sided “magnetic torque tunnel”, with each side having the same polarity, ensuring that all the magnetic fields are in the direction of motion and that there are no unused ends on the copper coils, wasting energy. All the magnetism the system creates is thus used to create motion, and all four sides of the stator contribute torque to the output. This allows smooth torque at slow speeds and then changes its operating patterns by grouping poles together as the motor speeds increase. This produces two to five times the torque density, at least three times the power density, and at least twice the total output of any permanent magnet motor of the same size. It also eliminates the need for DC/DC converters and gearboxes, reducing the total cost and weight of the vehicle significantly.

To improve the performance of electric motors, we need to effectively remove the heat from them. Equipmake’s solution is to rearrange the motor’s magnets so that they are positioned like the spokes of a wheel (Spoke motor). This not only increases the torque but also makes the magnets more accessible so that cooling water can be run directly over them. Recently, they have stated that with the addition of additive manufacturing, they intend to produce the world’s most power-dense permanent-magnet electric motor [161].

Yamaha Motor Co., Ltd., Shingai, Japan announced in February 2020 that it has begun the production of a high-performance electric-motor prototype that is capable of producing an industry-leading high-power density for automobiles and other types of vehicles. The compact unit generates up to 200 kW in output thanks to a high-efficiency segment conductor and advanced casting and processing technologies. It uses an interior permanent-magnet synchronous motor (IPMSM) with water or oil cooling [162].

The automotive supplier MAHLE has developed the most durable electric motor available. The traction motor, which is unique on the market, can run indefinitely with high performance. A new cooling concept makes this technological leap possible, as shown in Figure 23. The new electric motor is exceedingly clean, light, and efficient and can be assembled without the use of rare earths by customer request [163].

The new electric motor technology from Infinitum Electric is 50% lighter and smaller than traditional motors. It uses a so-called Air-Core PCB Stator, which replaces the heavy iron and copper components of conventional motors with a PCB stator to dramatically reduce size and weight (Figure 24). Here the copper coils are etched directly onto the PCB stator, allowing for a motor with a size and weight that is 50% less than traditional designs. Along with size and weight, other benefits of removing iron include reduced stator hysteresis and eddy current losses [164].

Koenigsegg’s new electric motor, named Quark (Figure 25) combines both radial- and axial-flux constructions to offer a good balance between power and torque. There is no need for a step-down transmission; instead, direct drive can be achieved, as the RPM of the motor is right from the start. Direct cooling was chosen for its higher cooling efficiency and compact design, while the rotor uses Koenigsegg Aircore™ hollow carbon fiber technology to make it lighter and smaller [165].

Israel-based EVR Motors Ltd., Petah Tikva, Israel created an entirely new topology, resulting in a completely new type of motor for the EV industry called the Trapezoidal Stator Radial Flux Permanent Magnet Motor, or TS-RFPM topology. The result is a smaller and lighter motor than the industry standard, which incorporates three-dimensional trapezoidal teeth and windings. Even more impressive, in some configurations, the TS-RFPM is rare-earth metal-free as they use ferrite magnets. They are also using soft magnetic composites (SMC) to improve performance as opposed to standard steel laminations. In addition to their high availability, high saturation and permeability levels, and lower eddy current losses, SMCs exhibit the necessary flexibility to tailor material performance to specific requirements [166].

BMW is going the other way. Its fifth-generation electric motor has no magnets. It operates as a three-phase AC synchronous motor using brushes and a commutator to provide power to the rotor windings. They even solved the problem of dust that comes with brushes rubbing the commutator by sealing them, making them more resistant to failure [167].

Another aim is to recycle electric motors, which for now are just being shredded and remelted. The problem is that, for now, there is no real interest in keeping the currently used electric motors as long as possible in operation. A study from 2021 [168] revealed that nearly half of the companies do not undertake any repair strategies for electrical machine components. Reusing EOL motors is, for now, not feasible because of a lack of information concerning the condition and availability of returned motors. Recycling focuses mostly on those metals that constitute a significant proportion of mass (for example, steel, aluminum, and copper), creating a risk that scarce metals are not being functionally recycled. There are EU projects that tackle this problem by repairing, remanufacturing, and reusing EOL electric motors, as well as developing new designs for the circular economy. The European project SUSMAGPRO has already presented some ideas for new designs [74]. Another European project, REASSERT [169], led by the Fraunhofer Institute for Manufacturing Engineering, will use AI tools developed as part of the project to help select the best value retention strategy for a given application. One of the goals is to develop a prototype motor for the circular economy that can be easily disassembled.

## 4. Future Perspectives

There is no future without electric motors. Whether there can be a future without using permanent magnets is questionable, but there will definitely be slower progress without them. Driven by the increased adoption of electric vehicles and green technologies, the global demand for Nd-Fe-B magnets (125,000 tones/year in 2019) is predicted to double to 250,000 t in 2030, as can be seen in Figure 26, with substantial growth in the automotive sector for the EV drivetrain. Figure 27 shows the shares of the different applications in the global Nd-Fe-B market. The analysis of applications shows that nearly 70% of magnets are used in applications based on either the rotational or linear principle of energy transformation from electric to mechanical—either as motors or as actuators. In addition, this percentage will grow in the coming years. With all the crises that the market for magnetic materials has already seen, it is not a question of if, but rather when the next crisis will occur. That is why research must go in new directions to increase the efficiency of electric motors with new designs, new magnet shapes and positions in the motor for higher fields, and a working circular economy to achieve values like alumina (more than 30%). In the following segment, we will analyze where the future of magnets and electric motors should be heading to achieve the goals needed for a successful transformation to a green economy and not letting monopolies control the market.

### 4.1. Designing the Future PM Electric Motor

The process of designing electric motors starts with defining the requirements presented in Figure 28. It is very important to compare the advantages and disadvantages of different types of electric motors.

To improve the overall mechanical efficiency of traction motors for the implementation of EVs, several issues need to be addressed. A study from 2024 [172] showed that, based on the design requirements identified for traction motors, Axial flux, in-wheel, and SRM have significant potential. Analyzing losses in traction motors reveals that speed and temperature are crucial parameters. The use of transmission technologies broadens motor efficiency and improves driving performance, with multi-stage Continuously Variable Transmission (CVT) and novel transmission systems expected to play a role. Efficient transmission system selection is not enough; layout optimization enhances traction motor and transmission efficiency. Independent wheel operation improves driving performance and overall efficiency, but cost and complexity must be managed. The choice of thermal management systems depends on the maximum operating requirements. Liquid cooling, especially for high-performance motors with stator cooling, is effective, while air cooling is preferable for rotor cooling to avoid cavitation. Heat enhancement technology can further reduce temperatures. Materials are crucial for mechanical and thermal performance, with novel materials being introduced to replace rare-earth metals and enhance traction motor performance. Advanced manufacturing (AM) technologies are evolving, impacting traction motor performance part-wise, but technological advancements are still needed on the topology front to match conventional production processes in density and price per unit volume [172].

To make the most efficient BLDC motor, there are various design options, as shown by the evolution of the rotor topologies with regards to the permanent magnets’ shape, mostly focusing on enabling radial orientation as opposed to diametral. Similar tendencies with shaped poles designed to gain the most gap flux and reduce spots with high demagnetization fields are common in wind generators. But if we take a close look at the schematic figures presenting the different types of electric motors in Figure 15, Figure 16, Figure 17, Figure 18, Figure 19, Figure 20, Figure 21, Figure 22, Figure 23, Figure 24 and Figure 25, we see that the magnets have uniform shapes, either rectangular or cylindrical. More complicated geometries, defined by the housing of the device, are usually realized by assembling smaller pieces together. For example, a circular rim is made of flat segments instead of a single curved piece, which results in reductions in the strength of the surrounding magnetic field. The loss must be compensated by using bigger segments, which limits the miniaturization and increases the price of the device. To a certain extent, the constructors of products must consider the standard shapes of the magnets. It is possible to apply post-processing machining to magnets, particularly to meet the additional requirements, for example, for heat-dissipation management. However, such operations, in terms of grinding or drilling, are undesirable because of the resulting waste and the possible negative impact on the performance of the magnet. It is obvious that the magnets of customized shapes, and consequently the magnetization, would contribute to the smaller size and improved performance of related devices. The recent emergence of additive manufacturing technologies could lead to devices like electric motors or electric generators being composed of magnets with a certain shape and distribution of the magnetic flux. It is assumed that the optimum performance of a device like an electric motor is only possible in the presence of a specially defined, tailored magnetic flux field (Figure 29) [173]. The goal is to associate the required magnetic flux field with its source in terms of the most suitable shape of the magnets. The latter is, together with the material properties, closely related to the magnetization distribution within the magnet.

The relationship between the magnetic flux field and magnetization is well defined by Maxwell’s equations [174]. The standard problem is to calculate the magnetic flux field, which is produced by the given magnetization as the source. Although the calculation might be demanding, the solution is uniquely defined, and the corresponding methodology is well established [175]. Magnet design requires the opposite approach, expressed in terms of the Inverse Magnetostatic Problem [176], to reconstruct the magnetization state of the magnet for a given magnetic field, which can be understood as an example of the reverse engineering of magnetic components. The main difficulties are related to the fact that the solution is not unique, i.e., there might be several magnetization distributions leading to the same field, of which the most appropriate should be found, or the solution might not even exist if the magnetic-flux field (requested, for example, by the electric-motor designer) is not valid according to the corresponding Maxwell’s equations. The modeling of magnets based on desired performance is tricky, and there is still a lot of room for improvement in the required algorithms and computer codes. An appropriate test of the method is a reconstruction of the magnetization in a Halbach cylinder that generates a homogenous field (Figure 30) [176]. Similar setups are directly applicable, for example, in the so-called Halbach motors.

### 4.2. Designing the Future Permanent Magnet

A more efficient electric motor requires higher magnetic fields. There are two ways to reach this goal. One is to increase the magnetic properties of the magnet by pushing them closer to the theoretical limit through chemical and microstructural engineering. The other is to construct the magnetic field by designing and positioning the magnets in a way that the electric motor makes the most of the field with new techniques, like additive manufacturing. This new design freedom will have a huge effect on how we see the electric motor, with possibilities we are not yet aware of.

#### Net-Shape Manufacturing

For new, complex designs, sintered magnets have to be cut, ground, and polished, which sometimes produces as much as 50% of waste. Net-shape production could eliminate this waste. One of the alternative approaches to manufacturing anisotropic, binder-less Nd-Fe-B magnets is spark-plasma sintering (SPS), which is a pulsed-current-activated, pressure-assisted technique. SPS can realize complex-shaped samples for materials such as ceramics, polymers, and alloys [177,178,179,180]. A quick survey of the existing literature shows that SPS has already been successfully adapted for the manufacture of binder-free magnets from different types of Nd-Fe-B powders (nanostructured melt-spun [181,182], gas-atomized [183], HDDR-type [184,185], and jet-milled [186,187], exploiting the rapid sintering kinetics and relatively low temperatures provided by this technique). The technique also offers flexibility in manufacturing multicomponent PMs, where the final magnetic properties of the magnet have different coercivities and remnant magnetization depending on the position in the magnet. This is created by the initial magnetic properties of the used constituents [188]. Such multicomponent magnets can be realized by using low- and high-HREE Nd-Fe-B powders. This is very practical in magnetic machines like wind turbines, for example, where it was predicted that a PM does not need to be equally resistant to demagnetization throughout the whole magnet volume. It was demonstrated that high coercivity is required mainly at the sides of a magnet, while the inner part can have a lower coercivity but a higher remnant magnetization [188]. Micro-magnetic modeling showed that magnetization reversal at elevated temperatures occurs on the surface, particularly at the corners and edges of the magnets/motors. Motor designers agree they mostly need high coercivity at these points (see Figure 31) [189].

It was found that the magnetic field around the Dy-free part is higher than in the Dy-rich part of the multicomponent magnets (see Figure 32) and that it is possible to arrange several magnetic entities in one magnetic body to profit from each region without suffering from unwanted effects. The possibilities for more complex designs of permanent magnets are thus open. On top of the best possible trade-offs in engineering, the magnetic properties of such manufactured multicomponent magnets exhibit an advantage as they reduce the price of the electrical machines in two ways: firstly, they require much less expensive and critical HREEs, and secondly, they lower the amount of other materials used to construct the machines that can now be more compact thanks to the higher remnant magnetization in the multicomponent magnet [188].

The second promising manufacturing route for net-shape PM production is additive manufacturing (AM). In the last decade, it has changed the industry as the next step in manufacturing. But for PM production, it becomes a very complex topic with many microstructural design constraints, but it can realize new design ideas with a net-shape production that could revolutionize the market (e.g., complex rotor core design, as schematically illustrated in Figure 33) [190]. With all the benefits that AM could bring, there are also enormous technical obstacles that still have to be resolved. There are many AM methods on the market to produce complex-shaped products. Concerning permanent magnetic materials, however, the currently available shaping methods are far from maturity. There were a few attempts to use AM for complex geometries using big-area additive manufacturing (BAAM) [191] or material extrusion (MEX) [192], which were demonstrated in a real-world use case, but because of the polymer dilution, only low remanences were achieved. A problem with MEX printing is also the lower dimensional accuracy and surface finish of the finished magnets compared to injection-molded magnets. SLA printers, which use stereolithography, exhibit a very high printing resolution (5 μm) compared to the other techniques but are limited to working well only with translucent materials because of the use of light for the polymerization. Injection-molding processes are one solution but require the creation of a mold, which is time-consuming and expensive, rendering this method impractical for producing small quantities. Binder jetting may be an alternative, where the green part is formed by compacting the powder with a binder that is sprayed onto the powder bed with ink-jet technology. Polymer-based metal 3D printing (Indirect AM) offers substantial design freedom: polymer filaments are filled with metal powder and printed on commercially available 3D printers. The polymer is removed via debinding, and the brown metal part is sintered to full density. This technique is already used to print stainless steel, copper, and other metal alloys. It can offer better control of the microstructure by employing existing sintering routines compared to laser- or electric-beam-assisted printing. However, polymer-based metal 3D printing processes face major challenges when printing Nd-Fe-B magnets. As shown in previous studies [193], polymer selection is crucial. Because of the high reactivity of the Nd-Fe-B powder, it can react with the carbon inside the polymer when de-binding, destroying any magnetic properties. Also, oxygen contamination plays a vital role in magnetic properties, which is why most of the production must be in a protective atmosphere. The same problem appears in NdFeB permanent magnets manufactured by Metal Injection Molding (MIM). While the green part has a better filling factor compared to 3D printed material extrusion parts, it suffers the same fate of high reactivity of the magnetic powder with organic elements. One is left with substantial residual carbon and oxygen contents, undermining their magnetic properties. To reduce the oxygen content, coating of the powder can be applied or the use of low-molecular-weight non-aqueous binder systems. Also, RE-rich alloys will better tolerate organic contamination. Metal Injection Molding of NdFeB magnets still presents numerous technical challenges but already produces magnetic parts with useful properties, which gives it a realistic processing route for permanent magnets [194].

In the field of powder bed fusion using a laser beam (PBF-LB), there have been improvements to produce fully dense metallic parts. The first tests showed initially promising results but were particularly challenging due to the pronounced peritectic in the Nd-Fe-B phase diagram [195,196]. This problem has been tackled by several other groups. Fine-tuning the laser melting parameters has a huge effect on the properties of the alloy [197,198]. Others showed that one can use grain boundary infiltration with low-melting-point eutectic alloys to further boost the coercivity of PBF-LB processed when infiltrated with Nd_50_Tb_20_Cu_30_, but with an additional processing step that impacts remanence [199]. Volegov et al. [200], using (NdPr)_3_Cu_0.25_Co_0.75_, achieved a higher room-temperature coercivity of approximately 1250 kA/m^1^. High coercivity can also be achieved by adding excess RE to synthesize over-stoichiometric powder compositions and a 2-step post-process annealing. Goll et al. [201] reached properties of *H*_CJ_ = 925 kA/m, *B*r = 0.58 T, and (*BH*)_max_ = 62.3 kJ/m^3^ with this technique. Tosoni et al. [202] used a copper-rich Nd-Fe-B composition synthesized close to industrial standards to reach even higher coercivities up to *H*_CJ_ = 1790 kA/m. This was achieved using a relatively low energy input during PBF-LB processing, which leads to extremely rapid cooling and hence a fine, equiaxed microstructure without dendrite growth or excessive α-Fe. Wu et al. showed that remelting Nd-Fe-B during PBF-LB led to the transformation of the coarse grains of the previously solidified layer to fine ones, favorable for the permanent magnetic properties [203]. Alignment of the magnetic particles in this type of AM is still a problematic task.

One of the main issues is the orientation of the printed magnetic particles. Most of the prints were isotropically oriented, which achieves only half of the potential remanence and only a quarter of the maximum energy product of such a magnet [204,205,206]. Producing anisotropic magnets is one of the major drawbacks of additive production. There have been some trial [207,208,209] printings of a permanent magnet or electromagnet, but as the print progresses, the magnetic field decreases, decreasing the orientation of the final printed product. Podmiljšak et al. showed that anisotropy is not a problem to achieve with a permanent magnet as a magnetic field source, but it has limitations in the dimensions it can provide [210]. Using an electromagnet for the field source improves the field strength, but this makes 3D printing less flexible [211]. Using post-aligning by heat-treating the sample in a magnetic field is another method that was used by Gandha et al. [209]. Sarkar et al. used a multiphysics model to simulate the alignment of magnetic particles in the presence of an externally applied field for an additively manufactured magnetic sample to help predict the magnetic properties of a 3D-printed part [212]. It is also possible to achieve alignment in PBF-LB samples. Goll et al. [213] produced a textured microstructure in Fe_73.8_-Pr_20.5_-Cu_2.0_-B_3.7_ alloy magnets through a non-rotating laser scanning strategy. This resulted in notable grain alignment along the laser scanning direction and an increase from 0.5 T (isotropic structure) to 0.67 T in remanence.

Another problem with different techniques of 3D printing is that the composition cannot be varied throughout the magnet. This is problematic with PBF-LB/SLM, binder jetting, and SLA printing techniques for printing magnets composed of distinct regions that are characterized by either a high intrinsic coercivity or a high remanent magnetization, i.e., multicomponent magnets, as presented in Figure 31. By placing an HRE-containing Nd-Fe-B powder only at those parts of the magnet that are at risk of being demagnetized during operation, such magnets can be designed for a particular motor application to reduce the consumption of HREs and boost performance. Novel dual- or multi-head FDM printers might be suitable for printing these kinds of magnets (Figure 34); however, the print-head change during the printing procedure is challenging for homogeneous printing results when a magnetic field is needed for anisotropic alignment. The final problem is that different thermal treatments are needed for multi-component materials. Accordingly, there is a need to develop advanced 3D printing technologies to preserve the PMs quality during the complete process, from the synthesis of the magnetic powder over filament fabrication to printing, de-binding, and sintering of the magnetic components.

## 5. Conclusions

The global economy is undergoing a transformative shift towards green electrification, a change driven by the urgent need to combat global warming. Numerous countries are rapidly moving away from internal combustion engines, setting ambitious targets for the adoption of electric and hybrid vehicles. For instance, Europe aims to achieve zero CO2 emissions from new cars and vans by 2035, aligning with its “Fit for 55” initiative [214]. In the United States, the goal is for half of all vehicles sold by 2030 to be electric or hybrid, sparking nearly $85 billion in investment into the electric vehicle (EV) industry over 2021 and 2022 [215]. China, leading the charge, mandates full electrification of new buses and urban logistics vehicles by 2025 and aims for all new passenger cars to be electric by 2035 [216]. This has propelled China to become a global leader in EV production and sales, with 6.8 million EVs sold in 2022 alone, dwarfing the U.S. sales of 800,000 EVs in the same period [217].

This electrification surge is not just a trend but a revolution, with carmakers releasing new electric models monthly and some planning a complete transition to electric drivetrains within a few years. This boom in production heightens the demand for raw materials, batteries, and electric motors, particularly magnets. Research is predominantly focused on developing more efficient batteries, improving by 10% annually, to address consumers’ range anxiety. However, this emphasis on batteries has led to a relative neglect of electric motor innovation. The design of permanent magnet (PM) electric motors, for instance, has seen modest changes over the past 50 years.

The rare earth element (REE) crisis has highlighted the vulnerabilities in the supply chain for permanent magnets, essential components in electric motors. These crises have spurred research and development in both permanent magnet materials and motor designs. Globally, efforts to diversify the supply of REEs are gaining momentum, with 146 advanced-stage REE projects, including new mines and existing operations, underway worldwide [218]. The opening of a new REE mine in Wyoming, the first in the U.S. in 70 years, is a significant development, signaling efforts to reduce reliance on foreign REE sources, predominantly from China [219].

However, achieving independence in the REE sector requires more than just raw material extraction. The construction of production facilities outside China, such as the $10 billion investment at the Mountain Pass mine, aims to establish a complete supply chain for magnet production [220]. Similar initiatives are underway in Europe, although they face ecological and economic challenges, including resistance from local communities.

Recycling is emerging as a crucial component of a sustainable REE supply chain. Projects like Europe’s Susmagpro have demonstrated the viability of large-scale recycling of permanent magnets. Yet the challenge of sourcing end-of-life magnets persists. To create a true circular economy for permanent magnets, there is a need for increased awareness among users and manufacturers, accompanied by legal requirements for recycling, labeling obligations, and recycling quotas.

In terms of material innovation, efforts to reduce the heavy and light REE content in PM magnets have been successful, but there is still no viable alternative to Nd-Fe-B magnets for high-power applications. Additive manufacturing offers new possibilities in magnet and motor design, but its current limitations in magnetic properties and mass production present obstacles to widespread adoption.

The future of green electrification is not limited to electricity alone. Innovations in alternative sustainable energy solutions, like Audi’s e-fuel [221], which produces e-diesel from wind energy, offer promising avenues. However, the success of such alternatives depends on the development of highly efficient PM generators for wind farms.

The shift toward green electrification is an unparalleled revolution, reshaping our world’s economic landscape and addressing the critical challenge of climate change. At the heart of this transformation is the need to optimize the performance and sustainability of permanent magnets, which are critical components in electric motors and generators. The pursuit of this goal involves a deep dive into the nanoscale intricacies of magnet materials, particularly the exploration of the microstructure-coercivity relationship in Nd-Fe-B magnets, which remains the gold standard for high-power applications.

Advanced research in material science is crucial for enhancing the energy efficiency and durability of these magnets. This includes investigating new alloy compositions, refining grain boundary engineering techniques, and exploring novel sintering processes to improve the thermal stability and corrosion resistance of magnets. Additionally, the development of alternative magnet materials that reduce or eliminate the reliance on rare earth elements is a key area of focus. Such materials need to match or surpass the performance characteristics of current REE-based magnets, particularly in terms of magnetic strength and temperature resilience.

Artificial intelligence (AI) and machine learning are emerging as pivotal tools in this domain [222]. By analyzing vast datasets encompassing material properties, manufacturing processes, and performance metrics, AI algorithms can uncover patterns and insights that elude traditional research methods. This approach can significantly accelerate the discovery of new magnet materials and the optimization of magnet designs, paving the way for more efficient and environmentally friendly electric motors and generators.

Furthermore, additive manufacturing (AM) technologies, such as Powder Bed Fusion-Laser Beam (PBF-LB), offer exciting opportunities for creating magnets with complex geometries and integrated cooling systems. These innovations could revolutionize motor designs, enabling more compact, efficient, and thermally stable electric motors. However, challenges in achieving the desired magnetic properties and scalability of AM-produced magnets must be addressed to realize their full potential in large-scale applications.

The journey towards green electrification is not just a matter of replacing fossil fuels with electric power. It is a scientific quest to push the boundaries of material science, physics, and engineering to develop sustainable, high-performance technologies that will drive the future of transportation, energy generation, and beyond. As we advance in this endeavor, the role of magnets becomes increasingly central, underscoring the need for continuous innovation and collaboration across disciplines to achieve a truly electrified and sustainable future.

## Figures and Tables

**Figure 1 materials-17-00848-f001:**
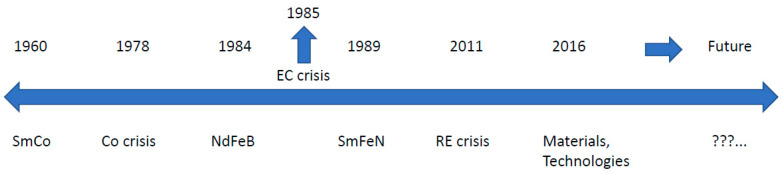
Time scale with triggering events and discoveries in the last 50-plus years.

**Figure 2 materials-17-00848-f002:**
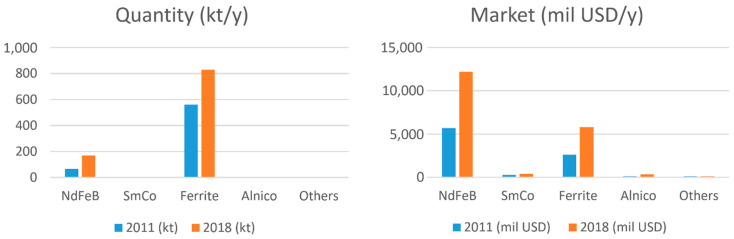
(**Left**) Permanent-magnet consumption by weight (2011 compared to 2018) in thousands of metric tons. (**Right**) Permanent-magnet usage in terms of annual turnover (2011 compared to 2018 in millions of US dollars per year) [10].

**Figure 3 materials-17-00848-f003:**
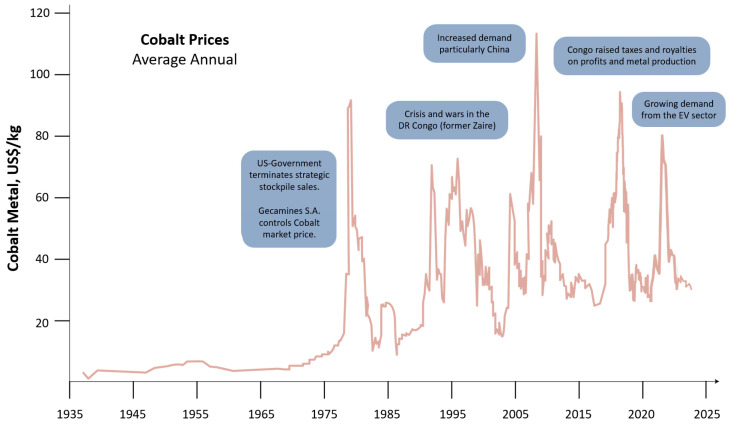
Average annual cobalt prices for the last 80 years.

**Figure 4 materials-17-00848-f004:**
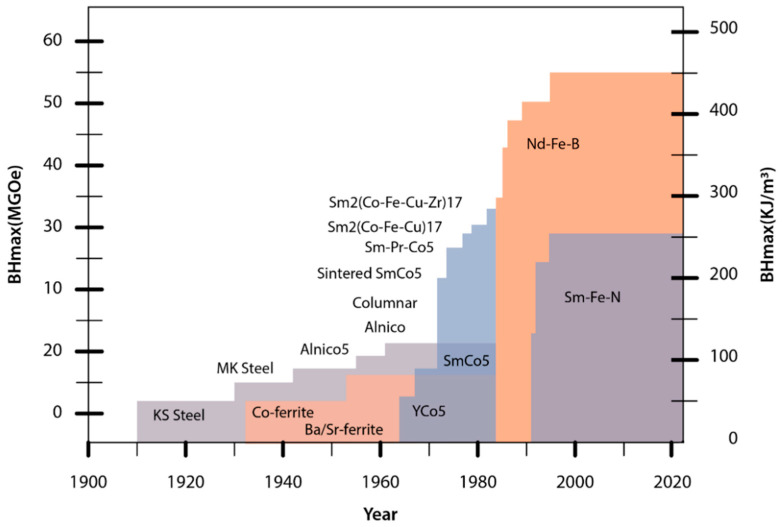
(*BH*)max values of different permanent magnets developed over the years.

**Figure 5 materials-17-00848-f005:**
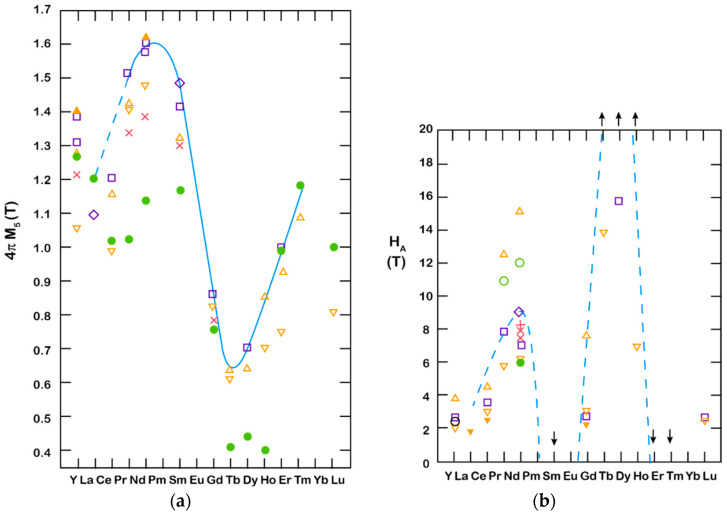
(**a**) Saturation magnetization vs. RE used in RE_2_Fe_14_B; (**b**) Anisotropy field vs. RE used in RE_2_Fe_14_B, reported by various authors. The symbols represent results from different authors. A detailed list can be found in Reference [21].

**Figure 6 materials-17-00848-f006:**
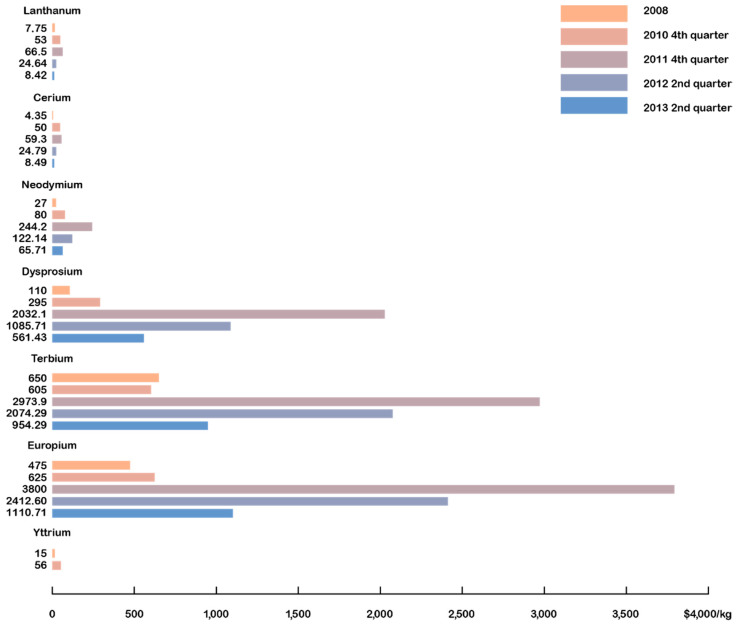
Selected rare-earth oxide prices, 2008–2013.

**Figure 7 materials-17-00848-f007:**
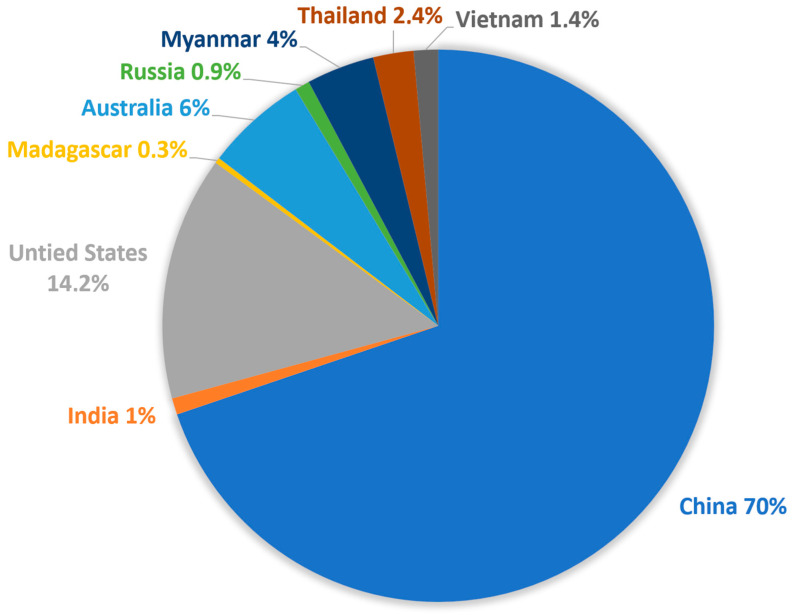
Rare earth metal production by country in 2022 [26].

**Figure 8 materials-17-00848-f008:**
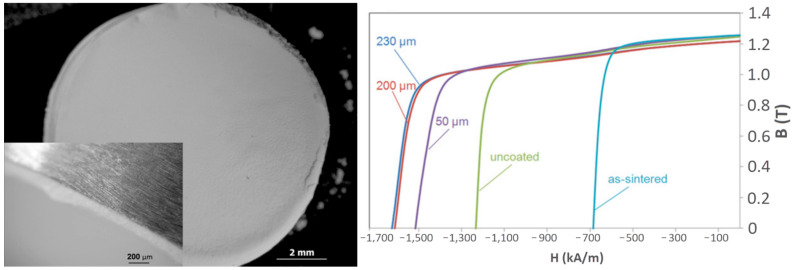
Enhancing the coercivity by deposition of HRE only on the grain surface [32].

**Figure 9 materials-17-00848-f009:**
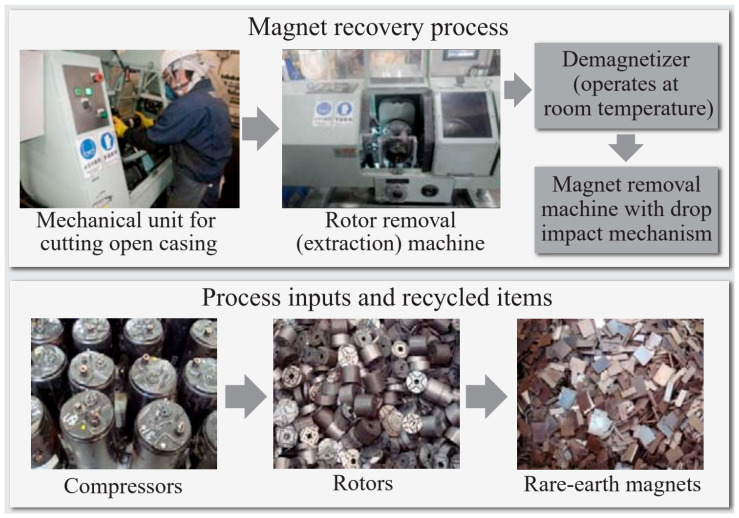
Magnet recovery process [48].

**Figure 10 materials-17-00848-f010:**
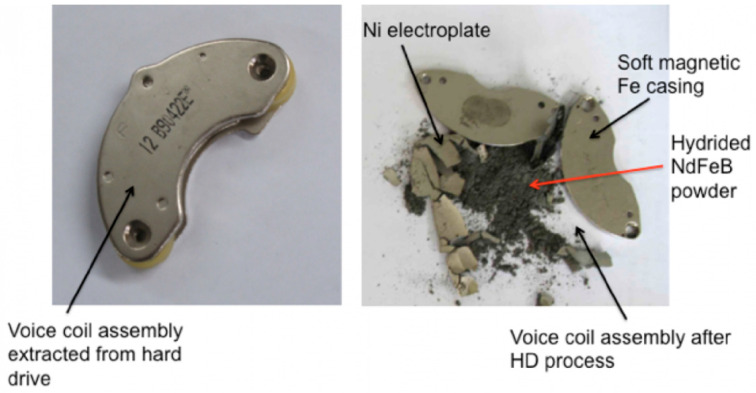
Illustration of the HPMS process applied to the voice-coil assembly of a hard disk drive [70].

**Figure 11 materials-17-00848-f011:**
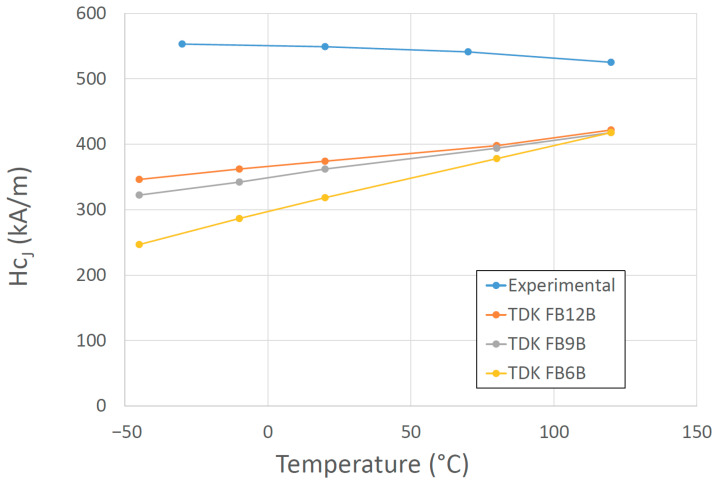
Temperature dependence of the intrinsic coercive force for a ferrite sintered magnet (Ca_0.5_La_0.5_Fe_9.1_Co_0.4_O_19-δ_) with 0.22 wt% B_2_O_3_ and several commercial magnets from the company TDK [80].

**Figure 12 materials-17-00848-f012:**
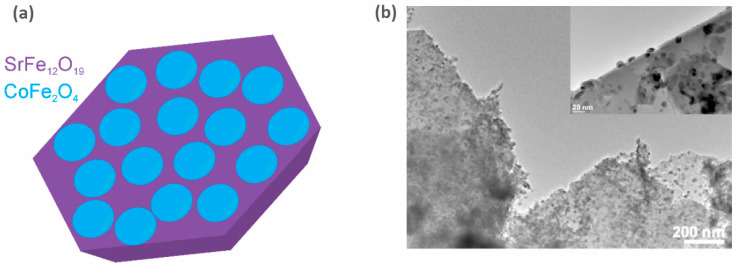
(**a**) schematic representation of CoFe_2_O_4_ nanoparticles distributed on the surface of a SrFe_12_O_19_ platelet and (**b**) TEM micrograph of the composite schematically presented in (**a**) prepared at the Department for Nanostructural Materials, Jozef Stefan Institute [106].

**Figure 13 materials-17-00848-f013:**
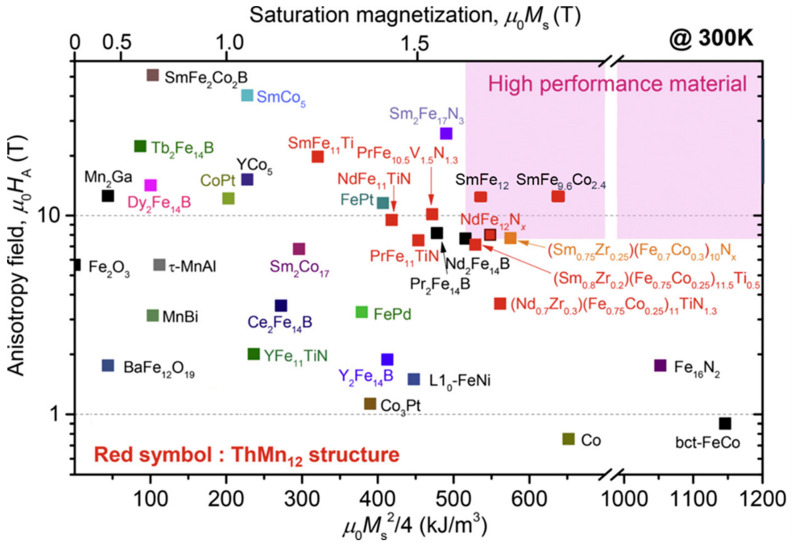
Anisotropy field (Ha) vs. theoretical (BH)_max_ for selected new magnetic materials [107].

**Figure 14 materials-17-00848-f014:**
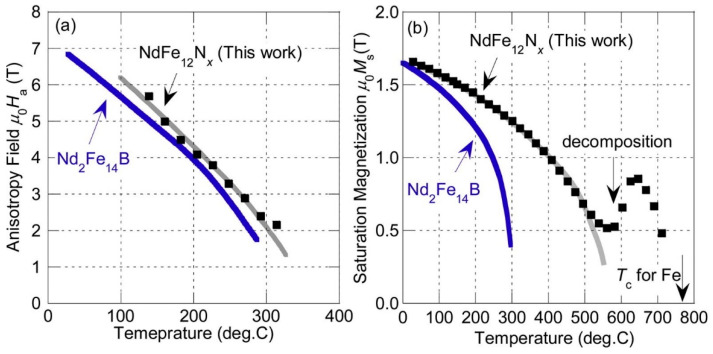
Anisotropy field (**a**) and magnetization (**b**) vs. temperature for NdFe_12_N [108].

**Figure 15 materials-17-00848-f015:**
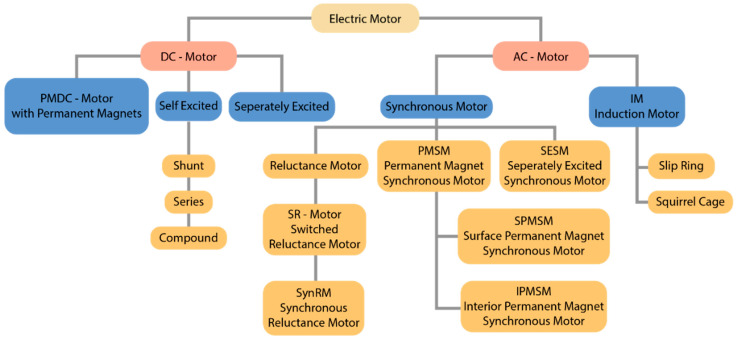
Overview of AC and DC motors.

**Figure 16 materials-17-00848-f016:**
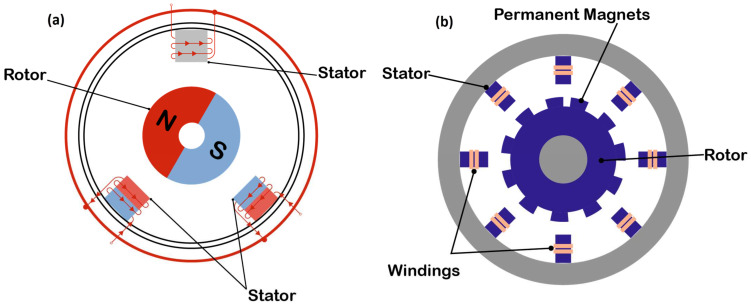
Schematic of a: (**a**) BLDC motor and (**b**) PMSM motor.

**Figure 17 materials-17-00848-f017:**
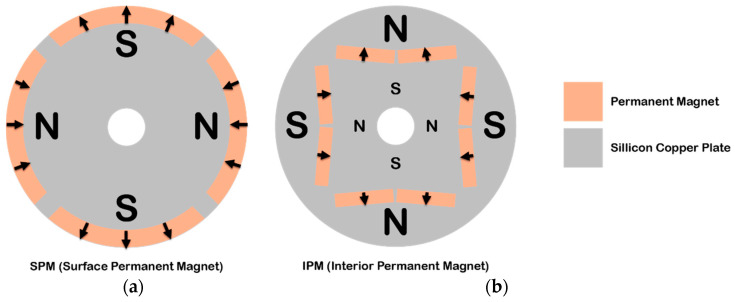
Examples of (**a**) SPM and (**b**) IPM motor rotor structures.

**Figure 18 materials-17-00848-f018:**
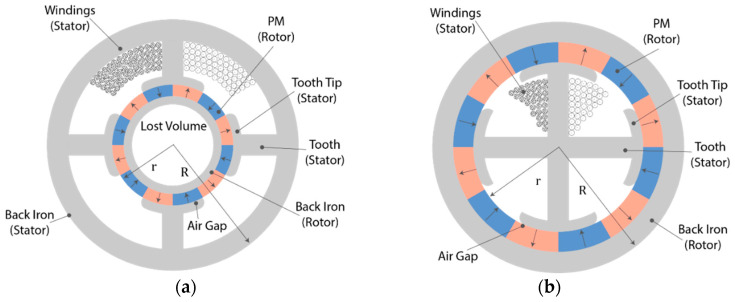
Compared to an internal rotor motor (**a**), an external rotor motor (**b**) has a larger area for the flux to develop and a larger air-gap radius, which acts as the “lever arm” for torque production [125].

**Figure 19 materials-17-00848-f019:**
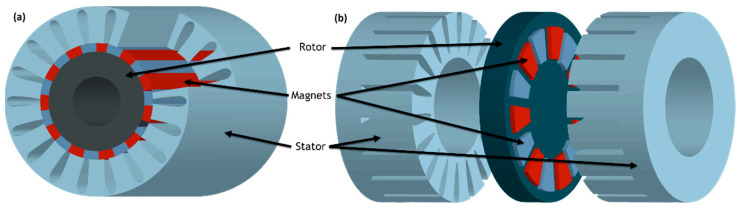
(**a**) the radial flux permanent-magnet synchronous machine (RF-PMSM) and (**b**) the axial flux permanent-magnet synchronous machine (AF-PMSM).

**Figure 20 materials-17-00848-f020:**
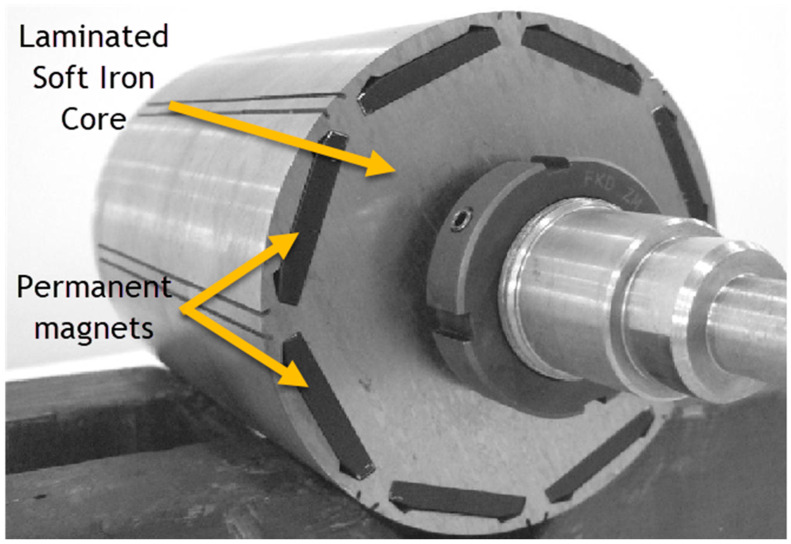
Rotor of a PMSM motor shows the laminated soft iron core [132].

**Figure 21 materials-17-00848-f021:**
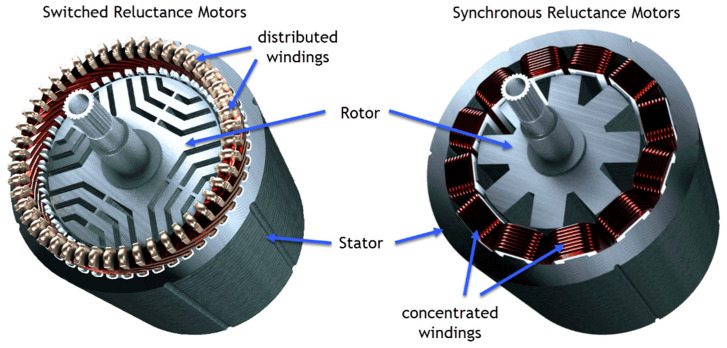
Synchronous Reluctance Motor [159].

**Figure 22 materials-17-00848-f022:**
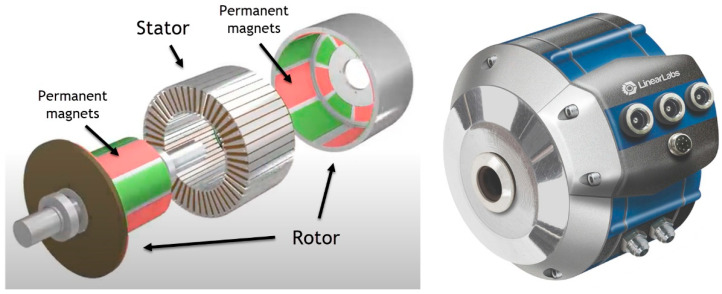
Hunstable Electric Turbine from Linear Labs.

**Figure 23 materials-17-00848-f023:**
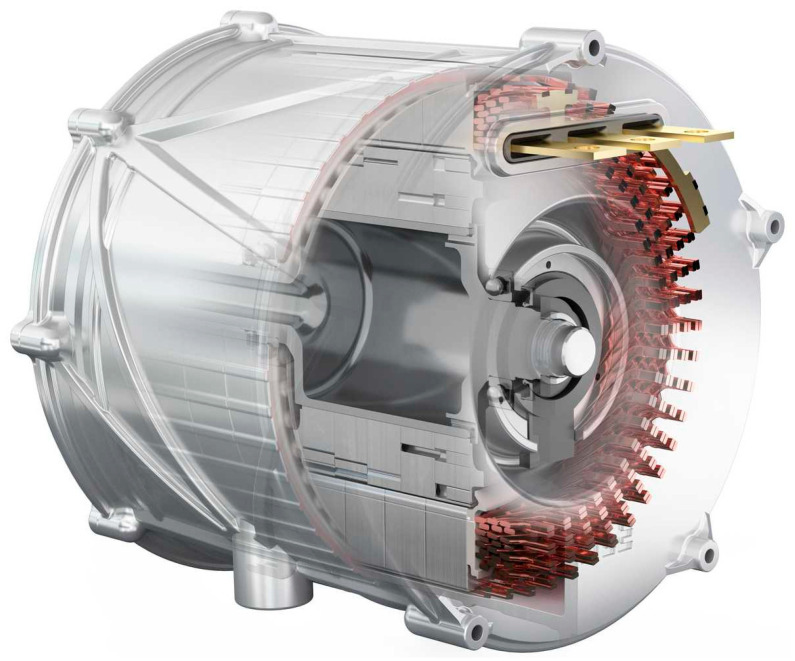
Mahle Superior Continous Torque (SCT) Electric Motor [163].

**Figure 24 materials-17-00848-f024:**
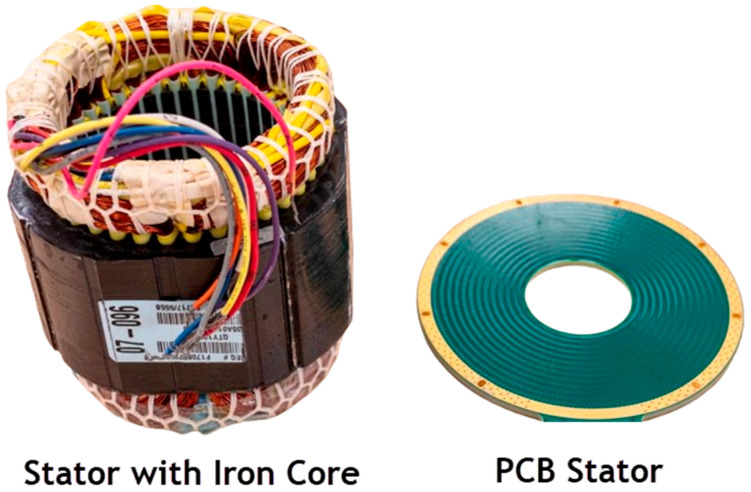
Comparison in size between the conventional Iron-copper component in an electric motor and the used PCB stator of the Infinitum Electric motor [164].

**Figure 25 materials-17-00848-f025:**
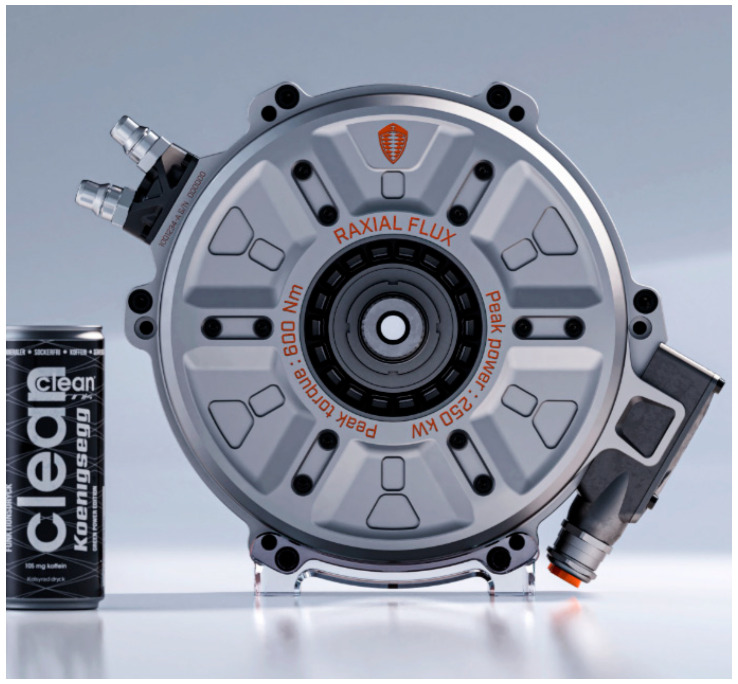
Koenigsegg Quark e-motor [165].

**Figure 26 materials-17-00848-f026:**
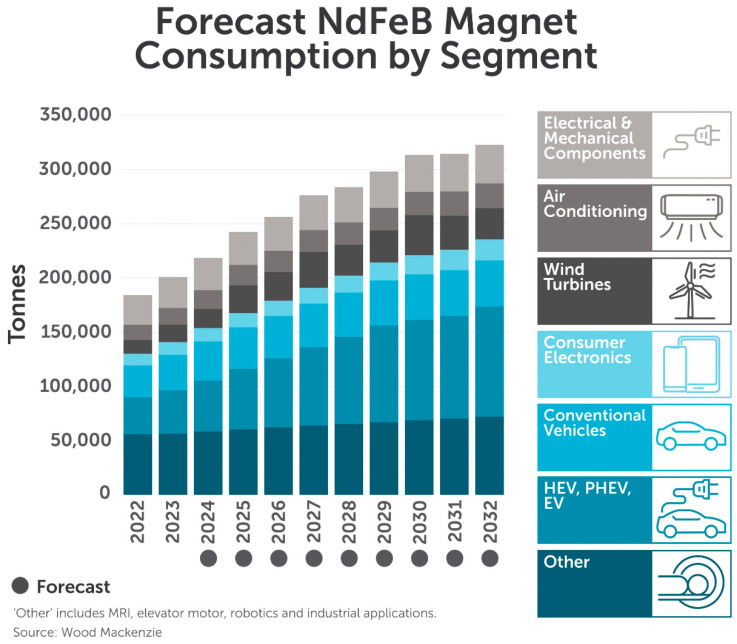
Current consumption of Nd-Fe-B magnets by applications and future predictions [170].

**Figure 27 materials-17-00848-f027:**
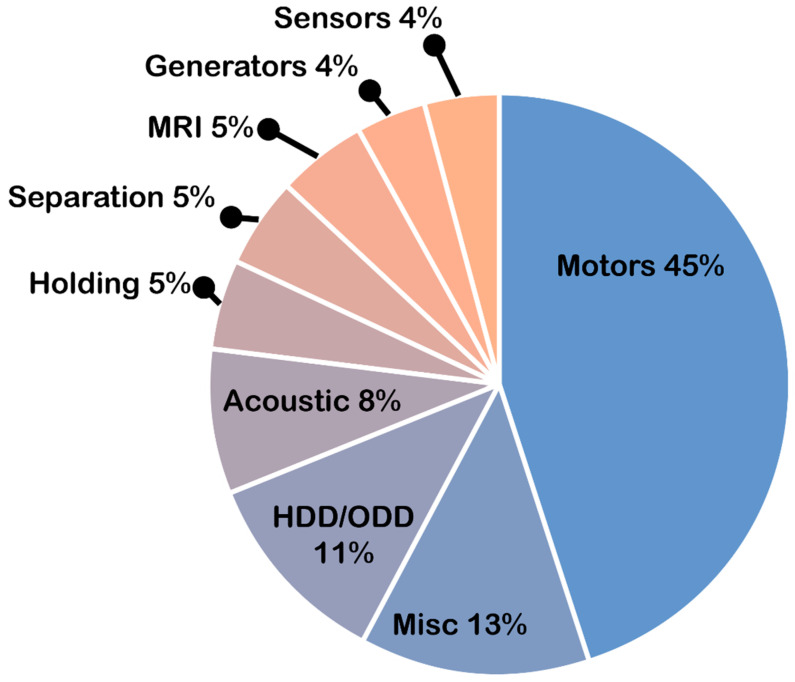
Use of permanent magnets in various applications (which fit into fields of motors, HDD/ODD, acoustic, and generators) [171].

**Figure 28 materials-17-00848-f028:**
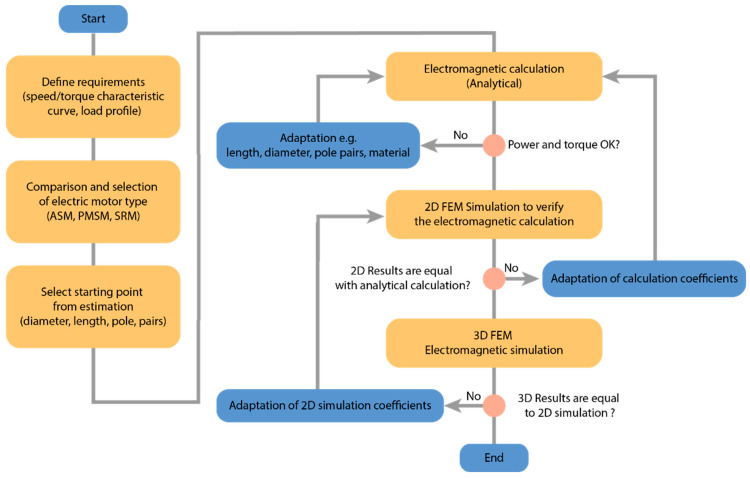
Process for designing electric motors.

**Figure 29 materials-17-00848-f029:**
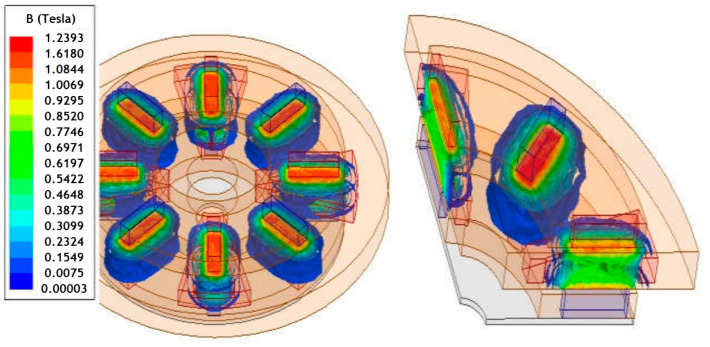
The force-line plot of the magnetic flux density in an electric motor [173].

**Figure 30 materials-17-00848-f030:**
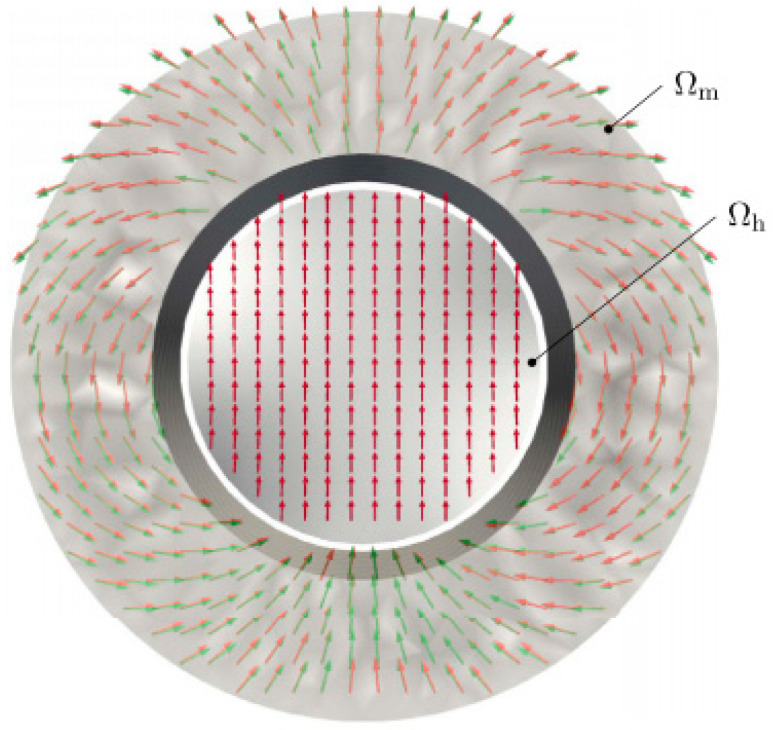
Reconstruction of a Halbach cylinder by means of an inverse magnetostatic problem. The straight red arrows denote a perfectly homogenous field in the inner region, whereas the tilted arrows demonstrate excellent agreement between the analytical (green) [176].

**Figure 31 materials-17-00848-f031:**
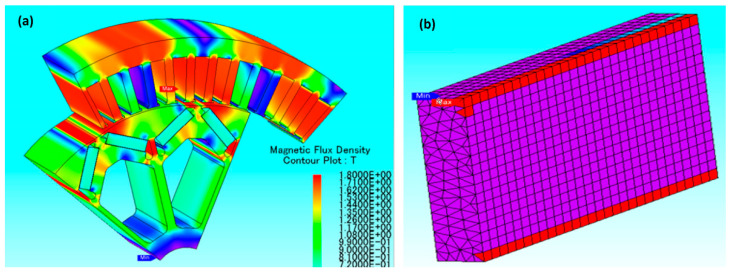
(**a**) Magnetic flux density distribution in the rotor/stator simulation and (**b**) demagnetization field simulated on a magnet from rotor (courtesy of JM Dubus, ROMEO project).

**Figure 32 materials-17-00848-f032:**
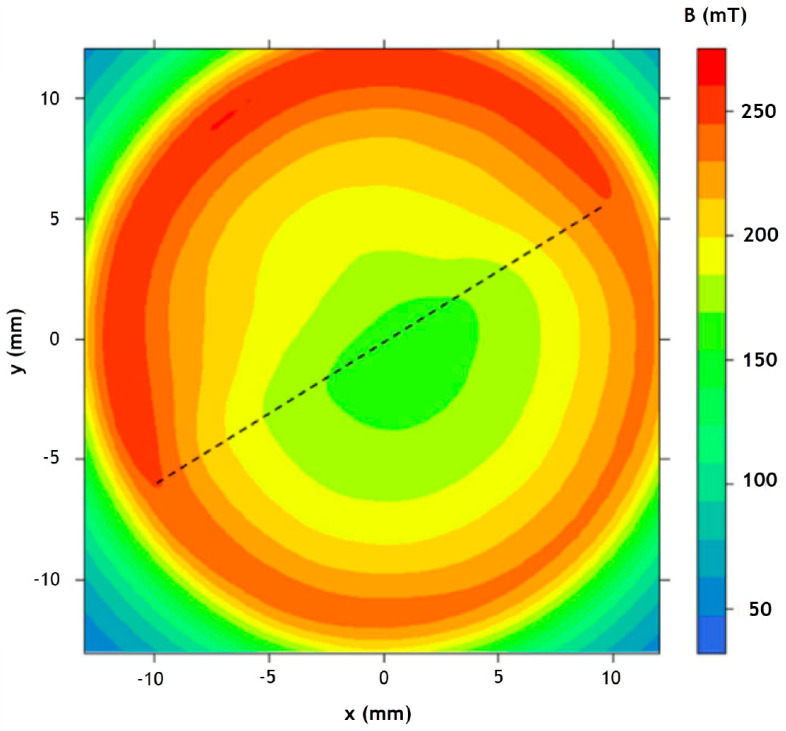
Absolute B-field measured 1 mm above the magnet’s surface with a Hall probe produced by SPS. The dashed line indicates the interface between the Dy-free (upper part) and the Dy-rich (lower part) materials [188].

**Figure 33 materials-17-00848-f033:**
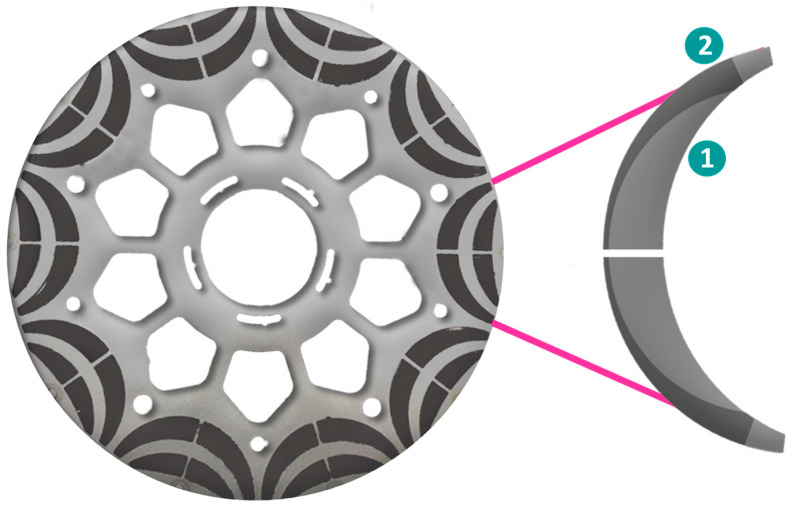
Example of an optimized magnet arrangement in an electrical motor with high coercive (1) and low coercive (2) material in the multi-component magnet.

**Figure 34 materials-17-00848-f034:**
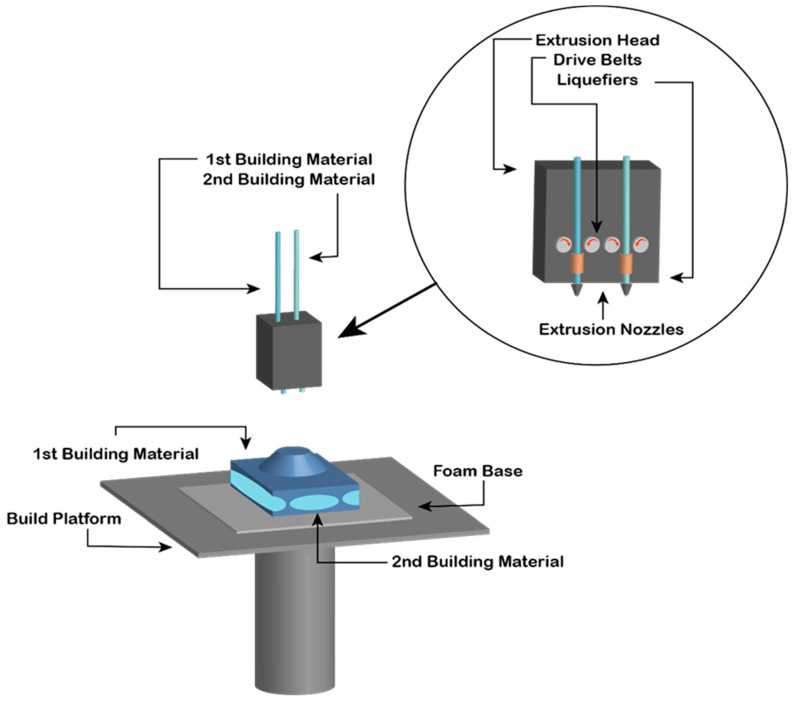
Schematic representation of a dual-head FDM printer with multi-material filaments.

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
