# Peer review of "The Future of Permanent-Magnet-Based Electric Motors: How Will Rare Earths Affect Electrification?"

_materials, 2024, doi:10.3390/ma17040848_

Round 1

Reviewer 1 Report

Comments and Suggestions for Authors

The research is scientifically and applied useful. It has a large volume of textual information, photographic material and computer simulations with concrete results. My rating is very high.

Need to fix:

- Slovenia ïn the line 6 "Jožef Stefan Institute, Department for Nanostructural Materials, Jamova cesta 39, 1000 Ljubljana, Sloveni"

- Despite the in-depth overview in the introduction, the data is only up to 2020 (Figure 3 and Figure 4). I recommend adding data until at least 2022 if possible.

- Similarly, more sources need to be included after 2020 - This is a recommendation, and I appreciate the already used over 150 sources.

I recommend making a detailed list of abbreviations from the main text and preparing it as an alphabetical glossary of terms. For example terms like: EV, REEs, TE, CEAM, HREs, GBPs, WEE and others should have a very short description. The article itself is long and thus will make it easier for StakeHodlers to read

Author Response

Dear Reviewer,

Thank you for your comments and recommendations. I made the suggested changes and added data up to 2024.

Best Regards,

Benjamin

Reviewer 2 Report

Comments and Suggestions for Authors

This article emphasizes the need for research and innovation in electric motor design, magnet materials, and alternative energy sources to achieve the goals of a fully electric transportation system. However, the below mentioned queries needs to be addressed from author side,

1)      The paper does not specifically address the current research gap in the field compared to current days research in the world.

2)      It does not provide a comprehensive analysis of the latest advancements, challenges, and gaps in the field of permanent-magnet-based electric motors.

3)      The paper does not delve into the specific limitations of existing electric motor designs or magnet materials.

4)      It does not discuss the potential limitations or drawbacks of additive manufacturing in electric motor design.

5)      The paper does not provide a detailed analysis of the impact of rare-earth elements on the future of electrification.

6)      It does not explore the limitations or challenges associated with developing a self-sustained circular economy for rare-earth materials.

7)      Please make sure to provide a Nomenclature.

8)      Please make sure to cite the most recent papers on the topic from high impact factor journals while providing a solid conclusion.

9)      Please make sure to provide a more solid conclusion with future recommendations.

10)   The level of English language should be checked throughout the manuscript and to make sure that the article is free from grammatical mistakes.

11)   Please emphasize to clearly indicate the novelty of your paper especially in the title, abstract and the introduction section. In the introduction section, please clearly indicate the research gap and the contribution that your paper will provide for the topic addressed in the field.

Comments on the Quality of English Language

Minor editing of english language required

Author Response

Dear Reviewer,

Thank you for your comments and recommendations. I tried to address all of your points as they are justified. I tried to update the review paper with new data and focused more on the problems that we are facing with a more solid conclusion. A more thorough check on the grammar was also made.

A Nomenclature is added.

Best Regards,

Benjamin

Reviewer 3 Report

Comments and Suggestions for Authors

The paper presents in a very comprehensive manner the evolution of the permanent magnet technology over the years, their relation to the electric motors used in electric vehicles and future perspectives focusing on manufacturing techniques. The manuscript fully aligns with the title playing not only the role of a review paper but also providing significant knowledge about the perspectives of REE for electrification.

There are a few minor points to be modified including also some mistakes of typographical nature that must be corrected:

1. Figures 5a and 5b should be explained a bit further as for exmple the symbols are not explained.

3. Line 144, the paragraph starts with "This The"

4. In line 213, "increased" should be used.

5. In line 229, "They used" instead of "The used"

6. In line 253, the word should be "environmentally"

7. In line 275, there is a double "it"

8. In lines 298 and 809, the subscripts and superscripts must be respected.

9. In line 326 the term "1 @m" is unclear.

10. In line 461, there is a single "o" i the beginning.

11. In section 4, page 19 Figure numbers are not correct.

12. Figure 24 does not have a reference

13. Title of subsection 4.1 (line 610) misses an "of "

14. In Figure 25, an arrow is missing leading from "Adaptation of 2D simulation coefficients" block to "2D FEM simulation ..." block

15. In line 616, it is BLDC and not BLDS.

16. In line 643, an "s" is missing form the word equation. It must be in plural.

17. The reference in Figure 26 must be 129 and not 126. Please check.

18. The reference (139) in line 692 is not in brackets.

19. All references to Figures from line 695 and on, seem to be downshifted by 4, i.e. 24 instead of 28 etc. Please correct.

Finally, some of the references are not appropriately written, like 6, 7, 10 for example, or 49, 50, 116, 135 (authors' names in capitals) and 71 (something is printed wrong).

Author Response

Dear Reviewer,

Thank you for your comments and recommendations. I tried to address all of your points as they are justified. I tried to check all the comments and corrected them appropriately.

I checked the reference for Figure 26 which you pointed out that it is incorrect, but I double-checked it and it is correct.

References are also updated to the correct standard.

Best Regards,

Benjamin

Reviewer 4 Report

Comments and Suggestions for Authors

The manuscript offers the holistic view on the topic of the influence of the rare earth element containing permanent magnet on the future electrification and electric transportation. A detailed overview on the historical and technical development of high performance hard-magnetic material is given in combination with market share, development of production volumes and application. This chapter is rounded up by a detailed overview on electric motor designs. To combine both topics, a design rule for specific motor applications is introduced showing up the points at which highest potential for improvements are identified. Besides a geometry optimisation, the cooling or local magnetic hardening of specific heat exposed magnetic parts is outlined. Obviously, the authors favour the use of Nd-Fe-B magnets due to the superior hard magnetic properties, which is concluded by a discussion of sustainability and accessibility of raw materials.     

Since this holistic view on the topic is very welcome and rarely found due to the high degree of specification of material scientists and electrical engineers, this review deserves publication to bring these topics together. However, there are some severe revisions requested to improve the publication quality significantly.

At first, the review literature data are lacking newest results – the latest citations are from 2020 as far as I observed. Please review literature to update possible research improvements. Especially in the field of additive manufacturing great leaps forward have been done during the last years. Alternatively the authors should quote directly that the review is back from 2020.

For example, highest magnetic performance Laser-PBF produced Nd-Fe-B magnets have been reported (O. Tosoni et al. 2023, Additive Manufacturing 64, 103426) or textured alignment of anisotropic magnet particles in polymer bond printed magnets (K. Schaefer et al. 2023, Journal of Magnetism and Magnetic Materials 583, 171064).

The method of Direct Energy Deposition (DED) is not considered at all for metal printing with a possible composition gradient in the material by mixing different feedstock powders in the printing nozzle. Multi-material LPBF is also available and leading to good results in printing of soft magnetic housings and rotors of electric motors.  

The authors also do not consider (even briefly) the role of the soft magnetic materials in the motor designs – are they not also relevant even for the permanent magnet containing motors?

More recently, also the Chinese car producers have obviously invested a lot in the development of efficient and cheap electric cars, so not only Volkswagen and Tesla should be considered in this review. Furthermore, electro-mobility Is not only cars, but also scooters and bikes have seen a huge increase in production numbers.

The conclusion that Nd-Fe-B based magnets will still be for the next years the material of choice for high performance electromobility and energy conversion is very probable and results directly from the literature research. However, the wording in the chapter is in my opinion chosen in a quite pathetic way. Please, for a scientific publication the political point of view should be considerably low, political decisions on material exports have clearly influenced market prices logically and hence enforced funding for scientific research. But political decisions (that have been proved independently that they were made) should not be referenced in a statement of a price war in a technological review in my opinion.

Further smaller points are:

1)      Fig 16: In the graph the macroscopic poles are given, but is it possible to show for clarification the magnetization directions of the embedded magnets?

2)      Some graphs (e.g. Fig 13) have very crowded axis value labellings or are rather not legible like in Fig 26, please improve the labelling.

3)      In several pictures of the electric motors from different companies (Fig 18 to Fig 22) the details in the pictures are not named (magnets, rotor, stator, coils…) and references of the pictures are not given in the caption. Please add the information for non-motor-designing experts.  

Comments on the Quality of English Language

Unfortunately, the general amount of typing errors and referencing errors makes the manuscript partly hard to read. Referencing errors occur mainly in the last chapters of the manuscript. Typing errors are all over making it hard to understand some formulas and abbreviations such as MS and HC (where consistent rules are available on correct writing of physical variables). The listed bullet points for several comparisons or summaries should be given as a table to increase the legibility.

Author Response

Dear Reviewer,

Thank you for your comprehensive review of my article. I agree with your comments and changed the article appropriately. I added newer references throughout the paper and also added sections that you suggested. I also rewrote the Conclusion as it was too political. I corrected also the images, except Figure 13 as it is taken as is from a published paper.

I also checked the grammatic errors and corrected them.

Thank you again for your comments.

Best Regards,

Benjamin

Round 2

Reviewer 2 Report

Comments and Suggestions for Authors

All the corrections suggested were implemented. Hence this can be proceeded to the next level of process